# A molecular atlas reveals the tri-sectional spinning mechanism of spider dragline silk

Wenbo Hu[1,3], Anqiang Jia[1,3], Sanyuan Ma[1], Guoqing Zhang[1], Zhaoyuan Wei[1], Fang Lu[1], Yongjiang Luo[1], Zhisheng Zhang [2], Jiahe Sun[1], Tianfang Yang[1], TingTing Xia[1], Qinhui Li[1], Ting Yao[1], Jiangyu Zheng[1], Zijie Jiang[1], Zehui Xu[1], Qingyou Xia[1] ✉ & Yi Wang [1] ✉

The process of natural silk production in the spider major ampullate (Ma) gland endows dragline silk with extraordinary mechanical properties and the potential for biomimetic applications. However, the precise genetic roles of the Ma gland during this process remain unknown. Here, we performed a systematic molecular atlas of dragline silk production through a high-quality genome assembly for the golden orb-weaving spider *Trichonephila clavata* and a multiomics approach to defining the Ma gland tri-sectional architecture: Tail, Sac, and Duct. We uncovered a hierarchical biosynthesis of spidroins, organic acids, lipids, and chitin in the sectionalized Ma gland dedicated to fine silk constitution. The ordered secretion of spidroins was achieved by the synergetic regulation of epigenetic and ceRNA signatures for genomic group-distributed spidroin genes. Single-cellular and spatial RNA profiling identified ten cell types with partitioned functional division determining the tri-sectional organization of the Ma gland. Convergence analysis and genetic manipulation further validated that this tri-sectional architecture of the silk gland was analogous across Arthropoda and inextricably linked with silk formation. Collectively, our study provides multidimensional data that significantly expand the knowledge of spider dragline silk generation and ultimately benefit innovation in spider-inspired fibers.

Spiders (Order Araneae) are abundant generalist arthropod predators including more than fifty thousand extant species[1,2]. All spiders produce silks, which are natural high-performance proteinaceous fibers that are crucial for spider survival and reproduction[3,4]. Silk production is a fascinating spider-salient trait of particular economic interest, mainly due to the exceptional properties of these fibers, including high tensile strength and toughness, low density, self-powered rotational actuation, and biocompatibility[5–7]. Numerous researchers have tried to emulate natural spidroin (the main proteins of spider silk) production and spinning processes for the biomimetic generation of artificial materials with spider silk-like properties[8–11]; however, much of our

current understanding of spider silk formation is based on physical and material studies that have provided only a partial picture of its nature[12,13]. Thus, the elucidation of the molecular biological mechanisms involved in the natural silk production system will be valuable for gaining an in-depth understanding of spider silk[14–17].

Although orb-web spiders have multiple silk-producing glands, the major ampullate (Ma) gland is often used as a model system in silk production research due to its relatively large size and, especially, the impressive properties of its product, dragline silk[18]. Accordingly, most attempts at innovating dragline silk-inspired fibers have generally involved the silk proteins and microenvironment produced by the Ma

[1]State Key Laboratory of Silkworm Genome Biology, Biological Science Research Center, Southwest University, Chongqing 400715, China. [2]School of Life Sciences, Southwest University, Chongqing 400715, China. [3]These authors contributed equally: Wenbo Hu, Anqiang Jia. ✉e-mail: xiaqy@swu.edu.cn; yiwang28@swu.edu.cn

gland[11,12,19]. The Ma gland can be divided into three macroscopic segments, the Tail, the Sac, and the Duct, which are characterized by gradients of pH values, ion concentrations, and shear forces[20–22]. Liquid silk protein is synthesized and stored at a very high concentration in the Tail and Sac and transformed into insoluble fiber via the Duct[23,24].

In this context, recombinant major ampullate spidroins (MaSps) have been constructed to achieve specific physical properties of silk, including strength, extensibility, and stickiness[25–27]. Spider silk-constituting elements (SpiCEs), which are nonspidroin proteins, have been utilized to increase the tensile strength in the case of composite silk films[28,29]. In addition, a microfluidic device designed to closely simulate natural ionic and pH conditions allowed fibers to be directly pulled from the outlet and then reeled in air, as in natural spinning[30–32]. It is becoming apparent that the detailed mechanisms underlying dragline silk production, including the cellular architecture and molecular function of the Ma gland as well as the biocomposition and formation of dragline silk, are fundamental to advanced fiber innovation[11,33].

To shed light on these mechanisms, we herein present a high-quality chromosome-scale reference genome for the golden orb-web spider *Trichonephila clavata*, which exhibits a colorful body and constructs a large and impressive orb web (Fig. 1a). By multiomic analysis of the Ma gland and dragline silk, we traced the origins of dragline silk components from the Ma gland segments (Tail, Sac, and Duct), elucidated the epigenetic and post-transcriptomic regulatory features of spidroin genes, and built a single-cell spatial architecture of the tri-sectional Ma gland, which provides the first detailed cytological definitions of the spider Ma gland based exclusively on segment-specific, cell-type-specific, space-specific, and dragline silk-related gene expression classification. These results allowed us to generate a comprehensive molecular atlas of natural silk production within the tri-sectional Ma gland, thereby elucidating the generation mechanism of dragline silk. The data were further extended to reveal the convergent evolution of silk production in the silkworm *Bombyx mori*, a model species from another arthropod group, and exhibited the shared molecular characteristics of the spider and silkworm silk glands. Our multiomics datasets are accessible in SpiderDB (https://spider.bioinfotoolkits.net) and will be valuable for future explorations of the evolutionary origins of silk production strategies and the creation of biomimetic spider silks.

## Results

### Chromosomal-scale genome assembly and full spidroin gene set of *T. clavata*

To explore dragline silk production in *T. clavata*, we sought to assemble a high-quality genome of this species. Thus, we first performed a cytogenetic analysis of *T. clavata* captured from the wild in Dali City, Yunnan Province, China, and found a chromosomal complement of $2n = 26$ in females and $2n = 24$ in males, comprising eleven pairs of autosomal elements and unpaired sex chromosomes ($X_1X_1X_2X_2$ in females and $X_1X_2$ in males) (Fig. 1a). Then, DNA from adult *T. clavata* was used to generate long-read (Oxford Nanopore Technologies (ONT)), short-read (Illumina), and Hi-C data (Supplementary Data 1). A total of 349.95 Gb of Nanopore reads, 199.55 Gb of Illumina reads, and ~438.41 Gb of Hi-C raw data were generated. Our sequential assembly approach (Supplementary Fig. 1c) resulted in a 2.63 Gb genome with a scaffold N50 of 202.09 Mb and a Benchmarking Universal Single-Copy Ortholog (BUSCO) genome completeness score of 93.70% (Table 1; Supplementary Data 3). Finally, the genome was assembled into 13 pseudochromosomes. Sex-specific Pool-Seq analysis of spiders indicated that Chr12 and Chr13 were sex chromosomes (Fig. 1b; Supplementary Fig. 2). Based on the MAKER2 pipeline[34] (Supplementary Fig. 1e), we annotated 37,607 protein-encoding gene models and predicted repetitive elements with a collective length of 1.42 Gb, accounting for 53.94% of the genome.

To identify *T. clavata* spidroin genes, we searched the annotated gene models for sequences similar to 443 published spidroins (Supplementary Data 6) and performed a phylogenetic analysis of the putative spidroin sequences for classification (Supplementary Fig. 12a). Based on the knowledge that a typical spidroin gene consists of a long repeat domain sandwiched between the nonrepetitive N/C-terminal domains[16], 128 nonrepetitive hits were primarily identified. These candidates were further validated and reconstructed using full-length transcript isoform sequencing (Iso-seq) and transcriptome sequencing (RNA-seq) data. We thus identified 28 spidroin genes, among which 26 were full-length (Supplementary Fig. 11a), including 9 *MaSps*, 5 *minor ampullate spidroins* (*MiSps*), 2 *flagelliform spidroins* (*FlSps*), 1 *tubuliform spidroin* (*TuSp*), 2 *aggregate spidroins* (*AgSp*), 1 *aciniform spidroin* (*AcSp*), 1 *pyriform spidroin* (*PySp*), and 5 other *spidroins*. This full set of spidroin genes was located across nine of the 13 *T. clavata* chromosomes. Interestingly, we found that the *MaSp1a−c* & *MaSp2e*, *MaSp2a−d*, and *MiSp-a−e* genes were distributed in three independent groups on chromosomes 4, 7, and 6, respectively (Fig. 1c). Notably, using the genomic data of another orb-weaving spider species, *Trichonephila antipodiana*[35], we identified homologous group distributions of spidroin genes on *T. antipodiana* chromosomes (Fig. 1d), which indicated the reliability of the grouping results of our study. When we compared the spidroin gene catalog of *T. clavata* and those of five other orb-web spider species with genomic data[28,29,36,37], we found that *T. clavata* and *Trichonephila clavipes* possessed the largest number of spidroin genes (28 genes in both species; Fig. 1e).

To further explore the expression of spidroin genes in different glands, all morphologically distinct glands (major and minor ampullate- (Ma and Mi), flagelliform- (Fl), tubuliform- (Tu), and aggregate (Ag) glands) were cleanly and separately dissected from adult female *T. clavata* spiders except for the aciniform and pyriform glands, which could not be cleanly separated because of their proximal anatomical locations and were therefore treated as a combined sample (aciniform & pyriform gland (Ac & Py)). After RNA sequencing of these silk glands, we performed expression clustering analysis of transcriptomic data and found that the Ma and Mi glands showed the closest relationship in terms of both morphological structure (Fig. 1g) and gene expression (Fig. 1f, h). We noted that the expression profiles of spidroin genes were largely consistent with their putative roles in the corresponding morphologically distinct silk glands; for example, *MaSp* expression was found in the Ma gland (Fig. 1h). However, some spidroin transcripts, such as *MiSps* and *TuSp*, were expressed in several silk glands (Fig. 1h). Unclassified spidroin genes, such as *Sp-GP-rich*, did not appear to show gland-specific expression (Fig. 1h).

In summary, the chromosomal-scale genome of *T. clavata* allowed us to obtain detailed structural and location information for all spidroin genes of this species. We also found a relatively diverse set of spidroin genes and a grouped distribution of *MaSps* and *MiSps* in *T. clavata*.

### Dragline silk origin and the functional character of the Ma gland segments

To further evaluate the detailed molecular characteristics of the Ma gland-mediated secretion of dragline silk, we performed integrated analyses of the transcriptomes of the three *T. clavata* Ma gland segments and the proteome and metabolome of *T. clavata* dragline silk (Fig. 2a). Sodium dodecyl sulfate−polyacrylamide gel electrophoresis (SDS−PAGE) analysis of dragline silk mainly showed a thick band above 240 kDa, suggesting a relatively small variety of total proteins (Fig. 2b). Subsequent liquid chromatography−mass spectrometry (LC−MS) analysis identified 28 proteins, including ten spidroins (nine MaSps and one MiSp) and 18 nonspidroin proteins (one glucose dehydrogenase (GDH), one mucin-19, one venom protein, and 15 SpiCEs of dragline silk (SpiCE-DS)) (Fig. 2b; Supplementary Data 10). Among these proteins, we found that the core protein components of dragline

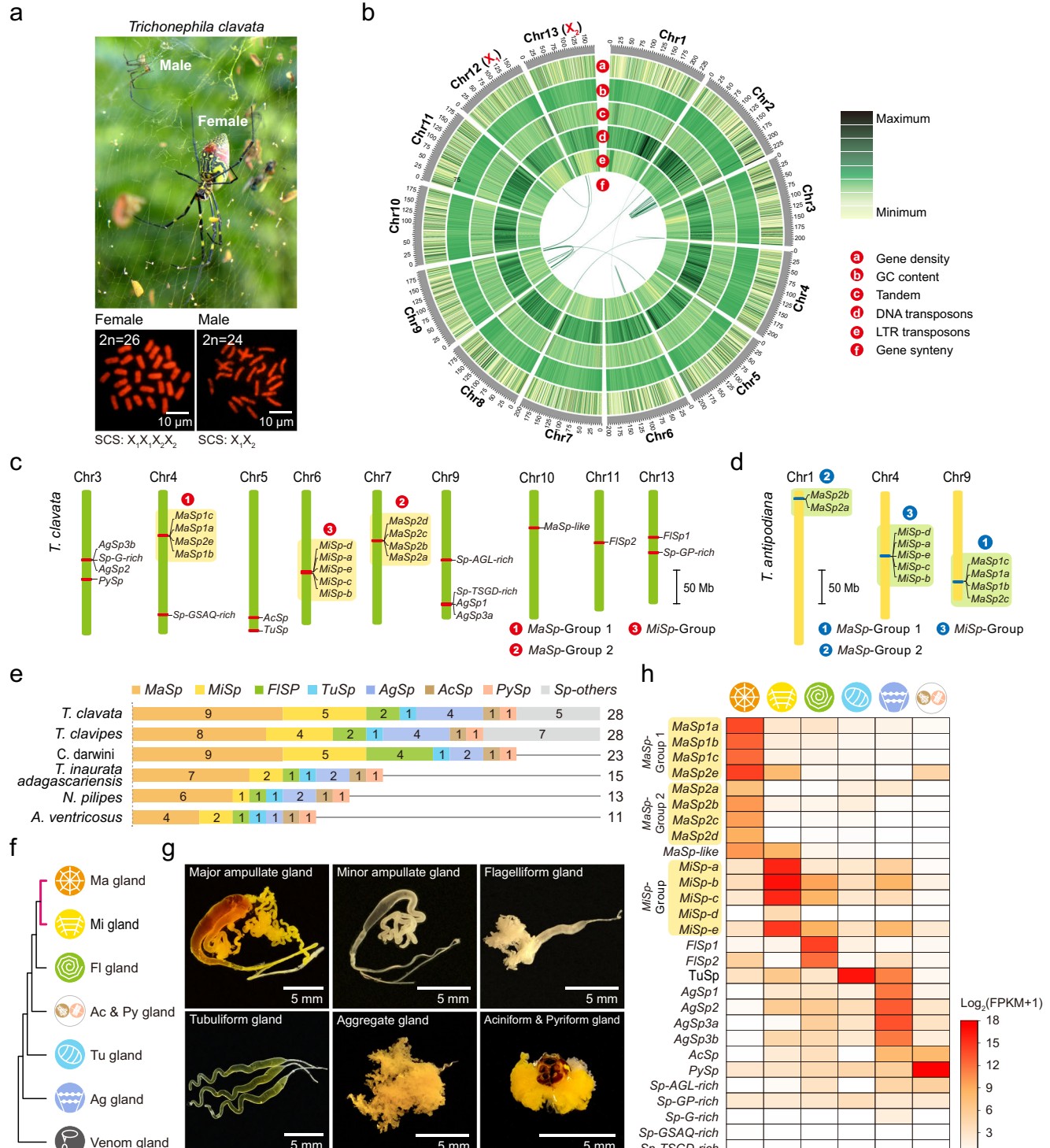

**Fig. 1 | Chromosomal-scale genome assembly and full spidroin gene set of *T. clavata*. a** Photograph of *T. clavata* showing an adult female and an adult male on the golden orb-web (above) and the female and male karyotypes (below). SCS, sex chromosome system. **b** Circular diagram depicting the genomic landscape of the 13 pseudochromosomes (Chr1–13 on an Mb scale). **c** Twenty-eight *T. clavata* spidroin genes anchored on chromosomes. **d** Spidroin gene groups of another orb-web spider, *T. antipodiana*. The published genomic data of *T. antipodiana*[35] was analyzed to identify the location information of spidroin genes. **e** Spidroin gene catalog of six orb-web spider species. **f** Expression clustering of silk glands (major and minor ampullate (Ma and Mi), flagelliform- (Fl), tubuliform- (Tu), aggregate- (Ag), and aciniform & pyriform (Ac & Py) glands) and venom glands. The pink line shows the closest relationship between the Ma and Mi glands. **g** Morphology of *T. clavata* silk glands. Similar results were obtained in three independent experiments and summarized in Source data. **h** Expression patterns of 28 spidroin genes in different types of silk glands. Source data are provided as a Source Data file.

**Table 1 | Characteristics of the *T. clavata* genome assembly**

| | |
|---|---|
| Estimated genome size (bp)[a] | 2,722,533,452 |
| Heterozygosity (%) | 1.44 |
| Length of genome assemble (bp) | 2,730,479,360 |
| Total length of scaffolds (bp) | 2,732,859,360 |
| N50 of scaffolds (bp) | 202,090,426 |
| N90 of scaffolds (bp) | 173,619,284 |
| Total length of contigs (bp) | 2,730,479,360 |
| N50 of contigs (bp) | 1,788,282 |
| GC content (%) | 31.80 |
| Complete BUSCOs[b] | 93.70 |
| Total length of pseudomolecules (bp) | 2,629,098,020 |
| Number of genes | 37,607 |
| Repeat content (%) | 53.94 |

[a]Genome size was estimated based on k-mer (GenomeScope).
[b]BUSCO analysis was based on metazoa lineage of protein-coding genes.

silk in order of intensity-based absolute quantification (iBAQ) percentages were MaSp1c (37.7%), MaSp1b (12.2%), SpiCE-DS1 (11.9%, also referred to as SpiCE-NMa1 in a previous study[28]), MaSp1a (10.4%), and MaSp-like (7.2%), accounting for approximately 80% of the total protein abundance in dragline silk (Fig. 2b). These results revealed potential protein components that might be highly correlated with the excellent strength and toughness of dragline silk.

To evaluate the composition of *T. clavata* dragline silk, we then assessed its metabolite composition and identified a total of 180 components (Supplementary Data 12). Among the metabolites, 109 were classified into ten categories: 34 organic acids, 22 organoheterocyclic compounds, 16 lipids, 13 benzenoids, 5 organic nitrogens, 8 organic oxygens, 5 nucleosides, 3 organooxygens, 2 phenylpropanoids and polyketides, and 1 alkaloid (Fig. 2c; Supplementary Data 13). We noted that xanthurenic acid (XA, a yellow pigment[38]) was the most abundant pigment (Fig. 2d), while other yellow pigments (such as carotenoids and flavonoids[39,40]) were not detected in our analysis, implying that XA is the major pigment providing *T. clavata* dragline silk with its golden coloration. The presence of XA was further confirmed by LC−MS analysis (Fig. 2e), consistent with a recent report[41].

To explore the origin of dragline silk components from the tri-sectional Ma gland, we focused on the transcriptomic features of the Tail, Sac, and Duct. We determined that the gene expression profiles of the Tail and Sac were more highly correlated with each other than with that of the Duct (Fig. 2f, g; Supplementary Fig. 13c), implying that the Tail and Sac have similar molecular functions. Furthermore, combined transcriptomic and proteomic analyses revealed the expression patterns of the 28 dragline silk protein transcripts in the Tail, Sac, and Duct (Fig. 2h; Supplementary Data 10). Notably, *MaSp1a–c* & *MaSp2e* (*MaSp*-Group1) were highly coexpressed in the Tail and Sac, and *MaSp2a–d* (*MaSp*-Group2) were highly coexpressed in only the Sac, while neither of these groups of proteins was highly coexpressed in the duct, indicating that the Tail and Sac are major silk-secreting segments.

We then used the tri-sectional Ma gland datasets to trace the source of the metabolite XA. We found that the genes encoding key enzymes involved in the XA biosynthesis (tryptophan metabolism) pathway were activated in all three Ma gland segments (Fig. 2i); in particular, a kynurenine aminotransferase gene (*KAT, Tc09G169510*) encoding the primary enzymes catalyzing the transamination of 3-hydroxy-L-kynurenine (3-HK) to XA showed this pattern. These findings suggested that XA is secreted by the Tail, Sac, and Duct.

To characterize the specific biological functions of the Tail, Sac, and Duct related to dragline silk production, we next assigned Gene Ontology (GO) terms to classify the functions of Ma gland segment-specific genes (Supplementary Data 14). We found that the GO terms that were significantly enriched in the Tail (relative to the Duct and Sac) were mainly related to the synthesis of organic acids (the largest group in the silk metabolome), those in the Sac (relative to the Duct and Tail) were mainly related to the synthesis of lipids (the third-largest group in the silk metabolome), and those in the Duct (relative to the Sac and Tail) were related to ion ($Ca^{2+}$ and $H^+$) exchange and chitin synthesis (Fig. 2j). Thus, a segmental division of biological functions was revealed.

Taken together, our results demonstrate a tri-sectional generation process of dragline silk in the Ma gland. Thus, we have established a genetic relationship between dragline silk components and the Tail, Sac, and Duct glands.

## Comprehensive epigenetic features and ceRNA network of the Ma gland tri-section

Based on the Ma gland RNA-seq data, we found that the total fragments per kilobase of transcript per million mapped reads (FPKM) of the dragline silk genes accounted for 47.49% and 34.33% of the FPKM values of the Tail and Sac, respectively; however, in the Duct, these genes accounted for only 0.76% of the FPKM values (Supplementary Data 11), indicating that the transcription of dragline silk genes was incredibly efficient in the first two segments. In particular, the *MaSps* within the two groups were highly coexpressed in the specific segments of the Ma gland (Fig. 2h). These findings revealed a segment-specific expression pattern of dragline silk genes.

To better understand the transcriptional regulatory mode of these genes, we first investigated genome-wide chromatin accessibility (CA) in the Tail, Sac, and Duct using the assay for transposase-accessible chromatin with sequencing (ATAC-seq). A total of 702,037 (Tail), 767,517 (Sac), and 653,361 (Duct) significant ATAC peaks (RPKM > 2) were identified in the 2 kb regions upstream and downstream of genes, and 10,501,151 (Tail), 11,356,55 (Sac), and 9,778,368 (Duct) significant ATAC peaks (RPKM > 2) were identified at the whole-genome level. The Tail (mean RPKM: 1.78) and Sac (mean RPKM: 2.04) plots showed genes with more accessible chromatin than the Duct (mean RPKM: 1.59) plots (Fig. 3a). We then analyzed the genome-wide DNA methylation level in the Tail, Sac, and Duct. We found the highest levels of DNA methylation in the CG context (beta value: 0.12 in Tail, 0.13 in Sac, and 0.10 in Duct) and only a small amount in the CHH (beta value: 0.04 in Tail, 0.05 in Sac, and 0.03 in Duct) and CHG (beta value: 0.04 in Tail, 0.05 in Sac, and 0.04 in Duct) contexts (Fig. 3b). Overall, there was no significant difference in methylation levels among the Tail, Sac, and Duct. Taken together, our results suggest a potential regulatory role of CA rather than DNA methylation in the transcription of dragline silk genes.

Next, the visualization of ATAC-seq and methylation datasets of the two *MaSp* groups in a genome browser revealed a reverse trend of peak signals (Fig. 3c, d; Supplementary Fig. 16). We analyzed potential TF motifs among the ATAC-seq peak sets in the 2 kb regions upstream of the transcriptional start sites (TSSs). We identified nine Tail- and Sac-specific TF motifs for *MaSp1b* (in *MaSp*-Group 1) and 13 Sac-specific TF motifs for *MaSp2b* (in *MaSp*-Group 2) (Supplementary Data 15; Supplementary Fig. 17a, b). Interestingly, we noted that the TF motifs closest to the TSSs, such as MYB and homeobox motifs for *MaSp*-Group 1 and two C2H2 motifs for *MaSp*-Group 2, were shared within each *MaSp* group (Fig. 3c, d). However, the Venn network of TF motifs between *MaSp*-Group1 and *MaSp*-Group2 showed little commonality among the Tail, Sac, and Duct (Fig. 3e). Therefore, we concluded that there was a common regulatory pattern within each *MaSp* group but a differentiated regulatory pattern between the two *MaSp* groups.

To investigate the impact of competing endogenous RNAs (ceRNAs: a post-transcriptional regulatory system implemented by miRNA and lncRNA[42]) corresponding to the regulation of dragline silk genes, we performed a whole-transcriptomic analysis of the tri-sectional Ma gland and identified a total of 527 miRNAs (179 in the Tail, 167 in the

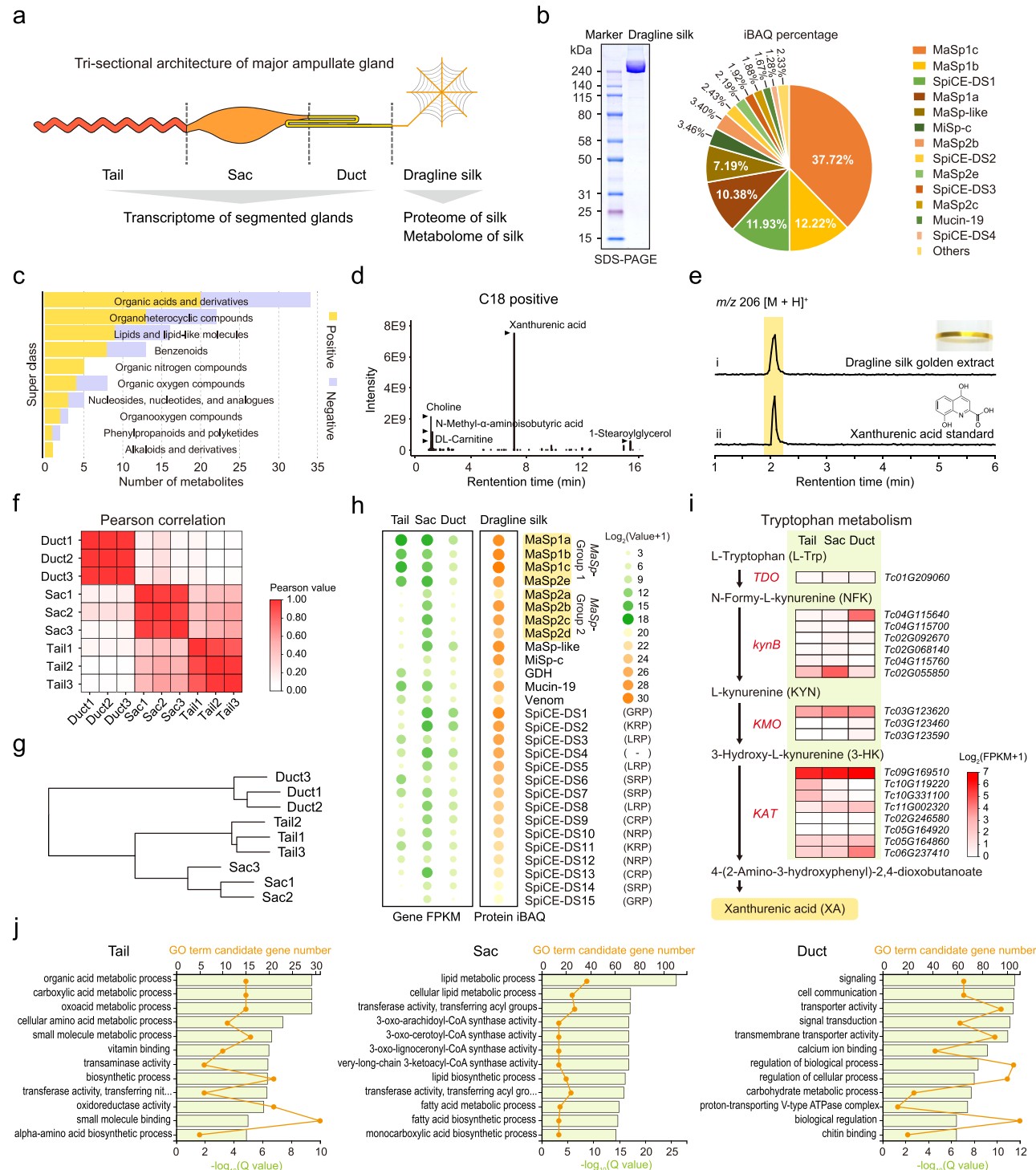

**Fig. 2 | Dragline silk origin and the functional character of the Ma gland segments. a** Schematic illustration of Ma gland segmentation. **b** Sodium dodecyl sulfate–polyacrylamide gel electrophoresis (SDS–PAGE) (left) and LC–MS (right) analyses of dragline silk protein. iBAQ, intensity-based absolute quantification. Similar results were obtained in three independent experiments and summarized in Source data. **c** Classification of the identified metabolites in dragline silk. **d** LC–MS analyses of the metabolites. **e** LC–MS analyses of the golden extract from *T. clavata* dragline silk. The golden pigment was extracted with 80% methanol. The extracted ion chromatograms (EICs) showed a peak at *m/z* 206 [M + H]⁺ for xanthurenic acid. **f** Pearson correlation of different Ma gland segments (Tail, Sac, and Duct). **g** Expression clustering of the Tail, Sac, and Duct. The transcriptomic data were clustered according to the hierarchical clustering (HC) method. **h** Combinational

analysis of the transcriptome and proteome showing the expression profile of the dragline silk genes in the Tail, Sac, and Duct. **i** Concise biosynthetic pathway of xanthurenic acid (tryptophan metabolism) in the *T. clavata* Ma gland. Gene expression levels mapped to tryptophan metabolism are shown in three segments of the Ma gland. Enzymes involved in the pathway are indicated in red, and the genes encoding the enzymes are shown beside them. **j** Gene Ontology (GO) enrichment analysis of Ma gland segment-specific genes indicating the biological functions of the Tail, Sac, and Duct. The top 12 significantly enriched GO terms are shown for each segment of the Ma gland. A *P*-value <0.05 was set as the criterion for screening significantly enriched GO terms. Source data are provided as a Source Data file.

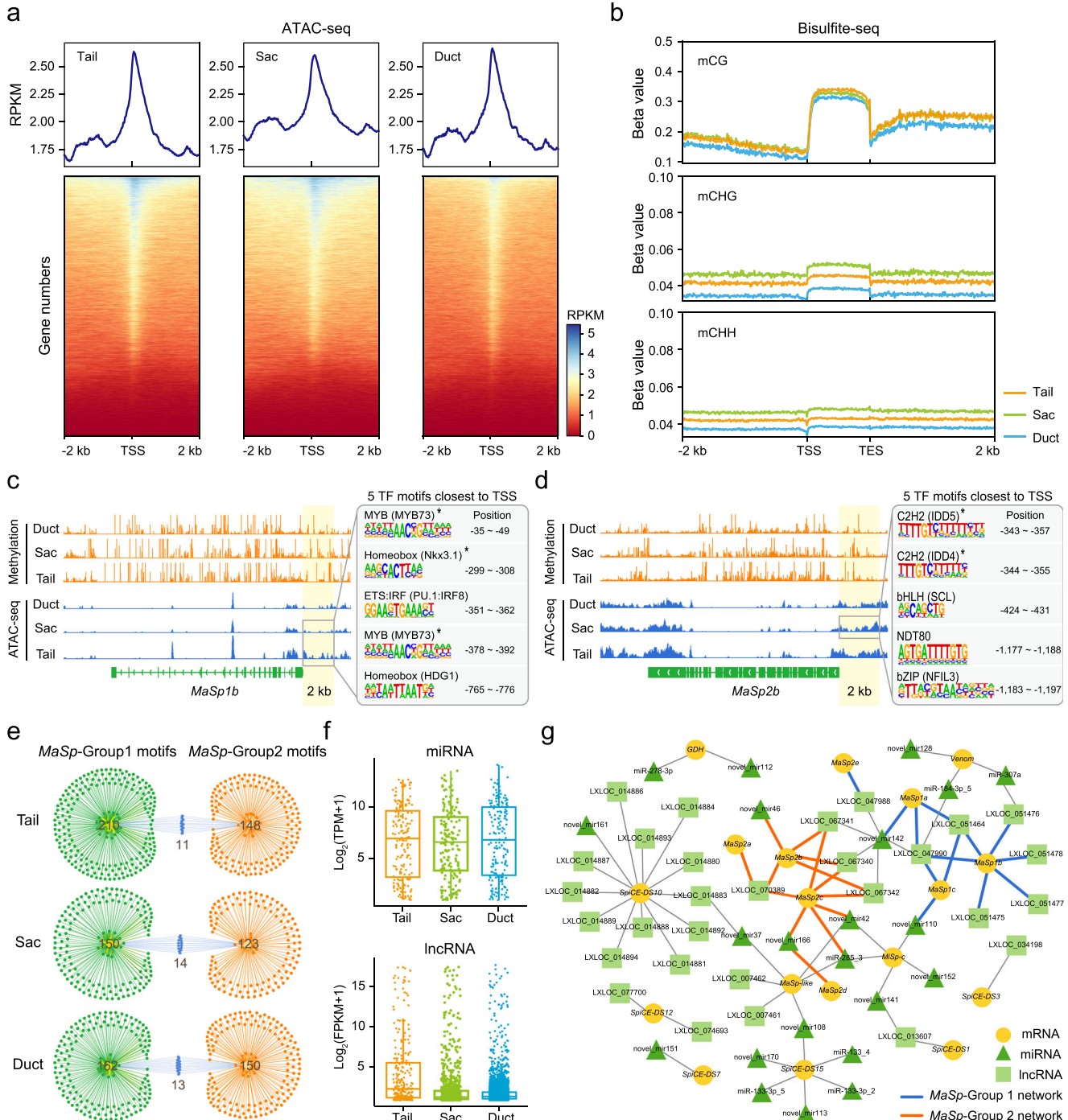

**Fig. 3 | Comprehensive epigenetic features and ceRNA network of the tri-sectional Ma gland. a** Metagene plot of ATAC-seq signals and heatmap of the ATAC-seq read densities in the Tail, Sac, and Duct. The chromatin accessibility was indicated by the mean RPKM value (upper) and the blue region (bottom). **b** Metagene plot of DNA methylation levels in CG/CHG/CHH contexts in the Tail, Sac, and Duct. (**c, d**) Screenshots of the methylation and ATAC-seq tracks of the *MaSp1b* (**c**) and *MaSp2b* (**d**) genes within the Tail, Sac, and Duct. The potential TF

motifs (*E*-value <1e$^{-10}$) in the indicated peak set (2 kb upstream of the TSS) are listed to the right and sorted by position. Asterisks represent the shared TF motif within the corresponding *MaSp* group. **e** Venn network of TF motifs between *MaSp*-Group1 and *MaSp*-Group2. **f** Expression levels of miRNAs and lncRNAs in the Tail, Sac, and Duct. Date are presented as mean ± SD (*n* = 3 for each Ma segment). Box plots show minimum to maximum (whiskers), 25–75% (box), median (band inside) with all data points. **g** ceRNA network of the dragline silk genes.

Sac, 181 in the Duct) and 10,110 lncRNAs (240 in the Tail, 982 in the Sac, and 4808 in the Duct) (Fig. 3f). From these data, we constructed a potential lncRNA–miRNA–mRNA interaction pairs by using the miRanda[43] and RNAhybrid[44] algorithms to identify the potential binding site between miRNA and lncRNA/mRNA, and then visualized the interaction networks by using Cytoscape software[45]. As shown in Fig. 3g, the ceRNA network of dragline silk genes consisted of 28

lncRNAs, 21 miRNAs, and 13 mRNAs. Remarkably, we noted that the ceRNA networks of *MaSp1a–c* & *MaSp2e* (*MaSp*-Group 1) were tightly clustered, as were those of *MaSp2a–d* (*MaSp*-Group 2); three lncRNAs (LXLOC_047988, LXLOC_047990, and LXLOC_051464) in the *MaSp*-Group 1 network were highly expressed in the Ma gland (Supplementary Fig. 18); one lncRNA (LXLOC_070389) and four miRNAs (novel_mir42, novel_mir46, novel_mir166, and miR-285_3) in the *MaSp*-Group

2 network were highly expressed in the Ma gland (Supplementary Fig. 18); in addition, the ceRNA networks of the two *MaSp* groups were independent of each other (Fig. 3g; Supplementary Fig. 18). These results further revealed potential post-transcriptional networks and the differentiated coregulatory pattern of the genes in the two *MaSp* groups in the Ma gland.

In summary, we observed an abundance of epigenetic and ceRNA signatures associated with the efficient and segment-specific transcription of dragline silk genes. Our data suggested the existence of differential regulation strategies in the three segments of the Ma gland dedicated to achieving the hierarchical gene expression of the *MaSp*s.

### Single-cell spatial architecture at the whole Ma gland scale

To further explore the cytological basis related to the hierarchical organization of the Ma gland, we generated single-cell and spatial patterns of gene expression in the *T. clavata* whole Ma gland. A total of 9349 high-quality single cells (SCs) were obtained after quality control, and they were then split into ten clusters through uniform manifold approximation and projection (UMAP) clustering (Fig. 4a). Based on the GO analysis of cluster-specific marker genes combined with the expression profiles of the segment-specific genes in each cluster (Supplementary Fig. 21 and Supplementary Note "Cell type annotation"), we carried out the fine annotations of ten SC clusters, namely, cluster 1: Ma gland origin cell (MaGO), cluster 2: MaSp-Group synthesis cell (MG1S), cluster 3: Chitin synthesis cell (CS), cluster 4: Unknown cell, cluster 5: Ampullate lumen skeleton cell (ALS), cluster 6: Ion transport cell (IT), cluster 7: Lipid synthesis cell (LS), cluster 8: pH adjustment cell (PA), cluster 9: MaSp-Group 2 synthesis cell I (MG2S I), and cluster 10: MaSp-Group 2 synthesis cell II (MG2S II), and delineated their sources from the Tail (clusters 1 and 2), Sac (clusters 1, 2, 5, 7, 9, 10), and Duct (clusters 1, 3, 4, 6, 8) (Fig. 4a).

To discern how the Tail, Sac, and Duct develop within the Ma gland, we ordered all single cells according to pseudotime and constructed a developmental trajectory. This resulted in a continuum of cells with three distinct branch points. We found that a set of cells from cluster 1 (ubiquitous SC cluster) assembled at the beginning of the pseudotime period and gradually bifurcated into four end-points representing two segments (the Sac and Duct) of the Ma gland, with the clusters arranged at different branch sites (Fig. 4b). We further investigated the developmental trajectories of the Tail, Sac, and Duct separately. As expected, most cells from cluster 1 were assembled at the beginning of the pseudotime period, while the Sac cells (clusters 2, 5, 7, 9, and 10) and the Duct cells (clusters 3, 4, 6, 8) were grouped into different branches (Fig. 4b; Supplementary Fig. 23). These results identified SC cluster 1 as the Ma gland origin cell and provided insights into the differentiation trajectories of Ma gland cells during cell state transitions.

We next assessed the spatial organization of cell populations in the Ma gland sections. This dataset contained gene expression information from a resource generated across 579 spatial transcriptomic (ST) spots within Ma gland sections (Fig. 4c). After analyzing the transcriptional signatures of ST spots, we identified seven spot clusters (one in the Tail, four in the Sac, and two in the Duct). As a first demonstration of the single-cellular and spatial gene expression patterns in the Ma gland, we visualized the expression of the *MaSp*-Group1 and *MaSp*-Group2 genes in UMAP and ST feature plots to better localize silk protein-secretion cells within our captured cell populations. Interestingly, we found that the *MaSp*-Group1 genes were prominently expressed in the Tail and Sac clusters (SC clusters 1, 2, 5, 7, 9, and 10 and ST clusters "a–f"), while the *MaSp*-Group2 genes were predominantly expressed in the Sac clusters (SC clusters 9–10 and ST cluster "d") (Fig. 4d; Supplementary Fig. 22a). Thus, the genes exhibited cellular and spatial cluster-specific patterns according to both the scRNA-seq and ST results that were consistent with the results of the expression and regulation analyses (Figs. 2h, 3c, d, g).

To characterize the identified SC and ST clusters related to the molecular function of the Ma gland, we selected 35 of the segment-specific genes identified in the segment transcriptomes to perform expression analyses based on the scRNA-seq and ST datasets. We found that SC cluster 2 and ST cluster "a" were major sets with the functions of organic acid metabolic process and oxidoreductase activity; SC cluster 7 and ST cluster "e" were major sets with the functions of lipid metabolic process and transferase activity; SC cluster 6 and ST cluster "f" were major sets with the function of calcium ion binding; SC cluster 3/8 and ST cluster "c/g" were major sets with the function of proton-transporting V-type ATPase complex; and SC cluster 3 and ST cluster "f" were major sets with the function of chitin binding (Fig. 4e; Supplementary Figs. 25, 26).

In summary, the detailed anatomic and molecular description of the Ma gland revealed a single cell type within the Tail and multiple cell types within the Sac and Duct, highlighting the developmental and functional differentiation of the Ma gland in the tri-section.

### Convergent evolution of the tri-sectional silk gland between *T. clavata* and *Bombyx mori*

To extend our investigation to an established model organism that has also evolved specialized glands for spinning silk, we chose the silkworm, *Bombyx mori*, which has a distinct phylogenetic position from spiders in the phylum Arthropoda[46]. Through anatomical observations, we noted a morphological convergence of the silk-spinning gland between the *T. clavata* Ma gland (Tail, Sac, and Duct) and the *B. mori* silk gland (posterior silk gland (PSG), middle silk gland (MSG), and anterior silk gland (ASG)), indicating a one-to-one correspondence of the tri-sectional architecture (Fig. 5a, b). We also found a similar number of silk gland cell types between *T. clavata* and *B. mori*[47] but differentiated annotations except for the chitin-related process in the Duct/ASG (Fig. 5b; Supplementary Fig. 29). Our previous studies revealed defects in silk spinning by silkworms caused by structural deficiency of the PSG and ASG[48,49]. However, the genetic manipulation system for spiders has not yet been established. To examine whether the remaining segment of the silkworm silk gland is required for proper silk production, we used *ser1* promoter-driven transgenic overexpression (OE) of a butterfly cytotoxin (pierisin-1A, P1A[50]) to generate an MSG-deficient silkworm (*P1A*-OE) (Fig. 5c). We found that the MSG of the *P1A*-OE strain was successfully truncated and that these silkworms failed to spin a cocoon (Fig. 5c), consistent with the phenotypes of PSG- and ASG-deficient silkworms[48,49]. Our results further demonstrated that the tri-sectional architecture of the silk gland is essential for silk spinning.

To further investigate the convergence between these two species, we performed whole-genome blastp alignment, and we identified 9593 and 7355 orthologous genes in *T. clavata* and *B. mori*, respectively. From these assigned gene ortholog pairs, we selected the *Hsp20* pair (*TcO4G175120* and *BMSK0007630*) because of the high sequence identity of 87.1% between *T. clavata* and *B. mori* (Fig. 5d); in addition, *Hsp20* was expressed in the silk glands of both *T. clavata* and *B. mori* and simultaneously served as a marker gene of SC cluster 5 of the *T. clavata* Ma gland (Supplementary Data 17), which encodes a small heat shock protein that acts as a protein chaperone to protect other proteins against misfolding and aggregation[51]. We next performed CRISPR/Cas9-based knockout (KO) of *Hsp20* in silkworm and successfully identified indels (96.46%) at the target site of the *Hsp20* sgRNA (Fig. 5d). Interestingly, we found that the silk production (cocoon weight and cocoon layer rate) of the *Hsp20*-KO strain was significantly lower than that of the wild type (Fig. 5e). From these results, we concluded that the orthologous *Hsp20* played a positive role in silk production in both the investigated spider and the silkworm.

To further investigate the molecular signatures of silk spinning for evidence of convergent evolution in the spider and silkworm, we next

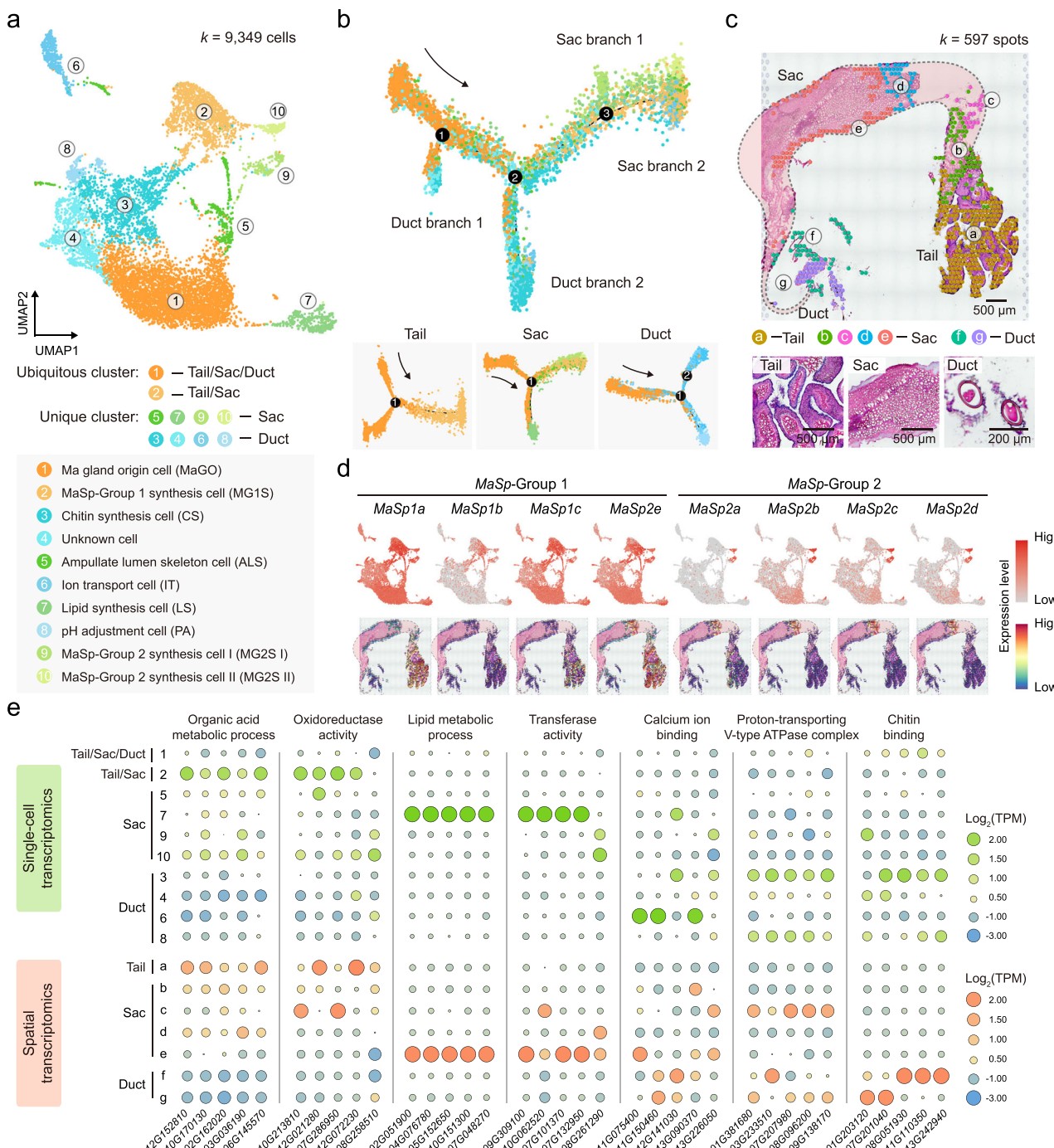

**Fig. 4 | Single-cell spatial architecture at the whole-Ma-gland scale. a** Uniform manifold approximation and projection (UMAP) analysis of cell types in the Ma gland and their grouping into ten cell clusters. The numbers in white-filled circles indicate cell clusters. The ubiquitous clusters are shown in the yellow series, the Sac clusters in the green series, and the Duct clusters in the blue series. **b** Pseudotime trajectory of all 9349 Ma gland cells. Each dot indicates a single cell, color-coded by the cluster as in (**a**). The numbers in black-filled circles indicate branch sites. The black arrows indicate the start of the trajectory. **c** Hematoxylin and eosin staining of Ma gland sections and unbiased clustering of spatial transcriptomic (ST) spots. Dotted lines depict the outline of the Ma gland. Similar results were obtained in three independent experiments and summarized in Source data. **d** UMAP and ST feature plots of the expression of genes in the *MaSp* group. **e** Heatmap showing the expression of Ma gland segment-specific genes in each cell type and each ST cluster along with corresponding GO terms. Source data are provided as a Source Data file.

performed comparative transcriptomic, proteomic, and metabolomic analyses of the tri-sectional silk glands and silks of *T. clavata* and *B. mori*. Shared GO terms were identified between each corresponding silk gland region of *T. clavata* and *B. mori*. We found that three GO terms (calcium ion binding, chitin binding, and signal transduction) were commonly enriched in the Duct and ASG and that one GO term (fatty acid metabolic process) was commonly enriched in the Sac and

MSG (Fig. 5f). These shared GO terms were silk gland-specific and not identified in other tissue types (hemocyte and ovary) (Supplementary Fig. 30). Next, we performed unique gene screening for each silk gland segment using a customized pipeline (Supplementary Fig. 28a). We found that 42, 6, and 2 pairs of orthologous genes were coexpressed in the Duct and ASG, Sac and MSG, and Tail and PSG, respectively (Fig. 5g; Supplementary Data 20). Interestingly, we found that seven ortholog

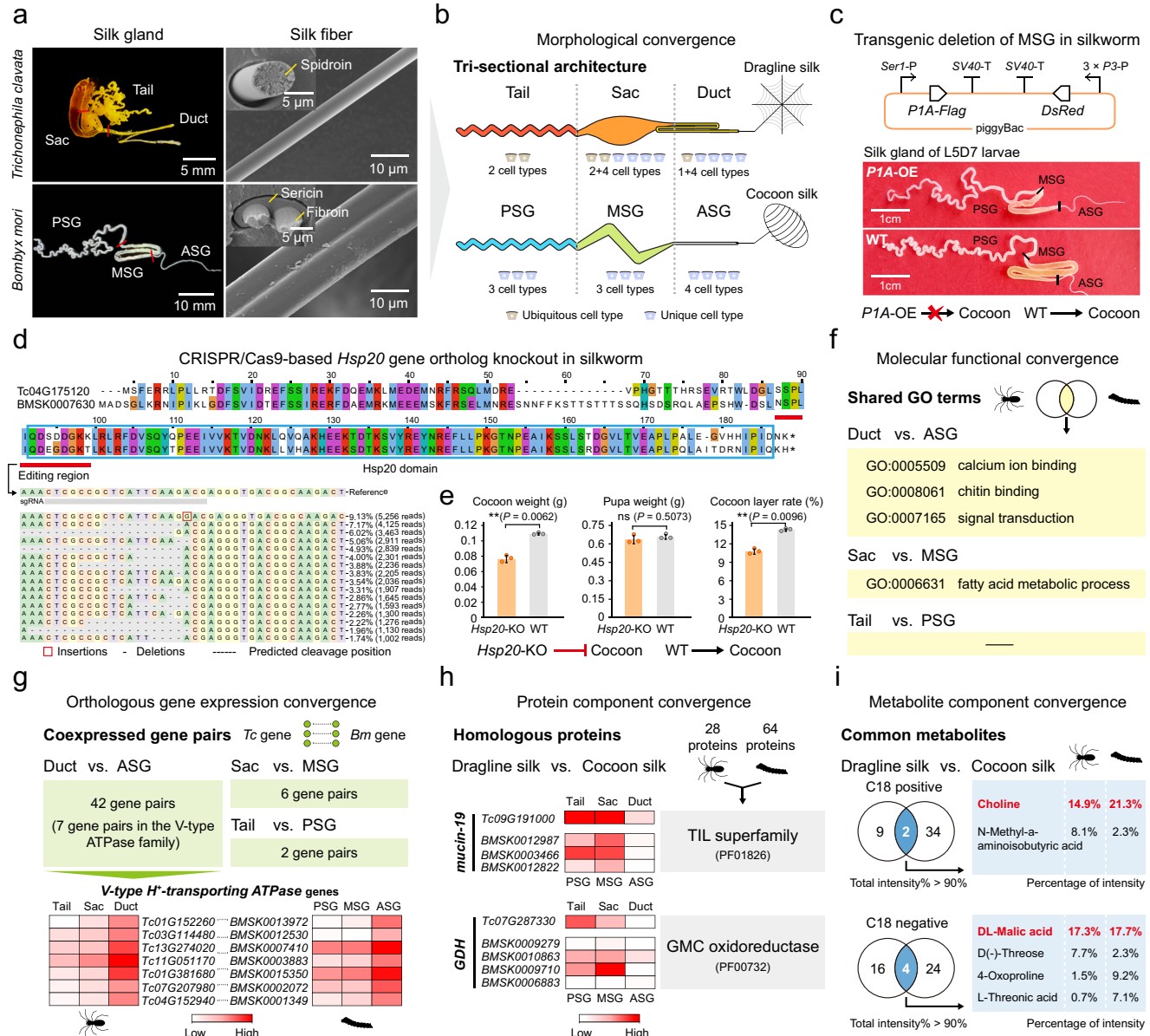

**Fig. 5 | Convergent evolution of the tri-sectional silk gland between *T. clavata* and *B. mori*. a** Morphology of the *T. clavata* Ma gland, dragline silk, *B. mori* silk gland, and cocoon silk. Insets show cross-sections of silk threads. Similar results were obtained in three independent experiments and summarized in Source data. **b** Schematic illustration showing the morphological convergence of the *T. clavata* Ma gland, with the Tail, Sac, and Duct indicated (above), and the *B. mori* silk gland, with the PSG, MSG, and ASG indicated (below). **c** Construction of the piggyBac transgenic vector (above) and silk gland phenotypes of the *P1A*-OE and wild-type (WT) strains in silkworm (below). *Ser1*-P, *Ser1* promoter. *3 × P3*-P, *3 × P3* promoter. *SV40*-T, *SV40* terminator. L5D7 represents the 7th day of the fifth instar. The arrows indicate the silk production process, and a red cross indicates that the process was blocked. **d** Sequence alignment of the orthologous Hsp20 proteins (Tc04G175120 and BMSK0007630) in spider and silkworm. The red line indicates the editing region. The arrow shows the distribution of identified alleles around the cleavage

site of the sgRNA. The top 16 sequences with a high percentage were exhibited. **e** Cocoon weight, pupa weight, and cocoon layer rate performance of the CRISPR/Cas9-based *Hsp20* knockout silkworm strain. Data were presented as mean ± SD (*n* = 3). Statistical comparisons were made using two-tailed Student's *t* test. ns indicates non-significant. **P-value <0.01. **f** Molecular functional convergence of the *T. clavata* Ma gland and the *B. mori* silk gland according to GO term analysis. A *P*-value < 0.05 was set as the criterion for screening significantly enriched GO terms. **g** Orthologous gene expression convergence between the *T. clavata* Ma gland and the *B. mori* silk gland. **h, i** Component convergence of protein (**h**) and metabolite (**i**) between silks produced by *T. clavata* and *B. mori*. The metabolites with a total intensity percentage above 90% were analyzed. The major metabolites in *T. clavata* dragline silk and *B. mori* cocoon silk are indicated in red. Source data are provided as a Source Data file.

gene pairs involved in the *V-type ATPase* family, encoding key factors involved in regulating silk fibrillogenesis[52], were significantly upregulated in both the Duct and ASG (Fig. 5g). Our results indicated a higher consistency between the Duct and ASG in molecular functions than between the Sac and MSG or the Tail and PSG.

We then explored whether the components of spider and silkworm silks showed convergence by performing Venn analyses of the

silk proteomes and metabolomes across the two species (Supplementary Fig. 28b). We found that mucin-19 and GDH were the only two proteins existing in both *T. clavata* dragline silk and *B. mori* cocoon silk, and more interestingly, these proteins were mainly synthesized and secreted by Tail/PSG and Sac/MSG (Fig. 5h). We also identified six common metabolites in these two silks: choline, N-methyl-α-aminoisobutyric acid, DL-malic acid, D(-)-threose, 4-oxoproline, and

L-threonic acid (Fig. 5i). Among these metabolites, it is worth noting that choline, a component of phospholipids in cell membranes[53], and DL-malic acid, which act as a preservative or pH-adjuster[54], were both major metabolites of dragline silk and cocoon silk (Fig. 5i). We therefore speculated that silk secretion from gland cells to the lumen is accompanied by choline release from the cell membrane and that, in natural silk, anti-rot and anti-bacteria characteristics are conferred by DL-malic acid.

In summary, the shared molecular characteristics of the tri-sectional silk glands of a spider and silkworm indicated convergent evolution under similar cases related to silk spinning and provided rich insights into silk gland function and silk biosynthesis.

## Discussion

Herein, we report the first chromosome-scale reference genome assembly of *T. clavata*. This high-quality genome combined with multiomics analyses enabled the elucidation of silk biosynthesis processes in the tri-sectional Ma gland. Our findings led us to hypothesize that the Tail performs the primary function in MaSps secretion, while the Sac, acting as a storage site, secretes a wide range of proteins (MaSps and nonspidroin proteins), and the Duct plays a limited role in protein secretion but a crucial role in protein structural transition[20,21,24].

We propose a molecular model comprehensively describing the mechanism underlying dragline silk biosynthesis in the *T. clavata* Ma gland (Fig. 6): (i) The Tail, consisting of two ubiquitous cell types, is the major site of the secretion of organic acids and 13 dragline silk proteins constituting the inner layer of dragline silk. The presence of organic acids can affect protein solubility in water by altering conformational states[55]. In the Tail, these 13 dragline silk genes are highly activated, among which the transcripts encoding *MaSp*-Group 1 proteins are the

dominant components. Relatively high CA in this segment, along with mCG methylation, might contribute to this abundant gene expression. (ii) The Sac, consisting of two ubiquitous and four unique cell types, is the major site of the secretion of lipids and 28 dragline silk proteins constituting the middle layer of dragline silk. The lipids act as a coat that wraps around the dragline silk to regulate its water content[55]. These 28 dragline silk genes are also highly activated in the Sac, among which *MaSp*-Group1, *MaSp*-Group2, *MaSp-like*, *SpiCE-DS1*, *SpiCE-DS2*, *SpiCE-DS4*, and *SpiCE-DS13* are the dominant components. Relatively high CA and mCG methylation might also contribute to the high gene expression observed in this segment. (iii) The Duct, consisting of one ubiquitous and four unique cell types, is the major site of the secretion of chitin, cuticular proteins, ions ($Ca^{2+}$ and $H^+$), and 13 dragline silk proteins. Chitin and cuticular proteins form the cuticular intima, where shear forces are generated[21,56]. The ions are responsible for a multi-dimensional state of flux, including ion exchange, pH gradient formation, and dehydration[11,20], and the 13 dragline silk proteins constitute the outer layer of dragline silk. The transcription of dragline silk genes is less active in the Duct than in the Tail and Sac, and only 0.76% of the transcripts come from these genes. Relatively low CA in this segment, along with mCG methylation, might contribute to this decrease in expression. Overall, our results not only define the proteomic and metabolic components of dragline silk but also trace their origins from the tri-sectional Ma gland. The numbers of cell types in the Tail, Sac, and Duct are positively correlated with the diversity of their functions.

Our results provide genomic clues for the hierarchically ordered biosynthesis of spidroins. We documented that the *MaSp1a−c* & *MaSp2e*, *MaSp2a−d*, and *MiSp-a−e* genes are distributed in three distinct groups. In addition, we demonstrated that the *MaSps* within each

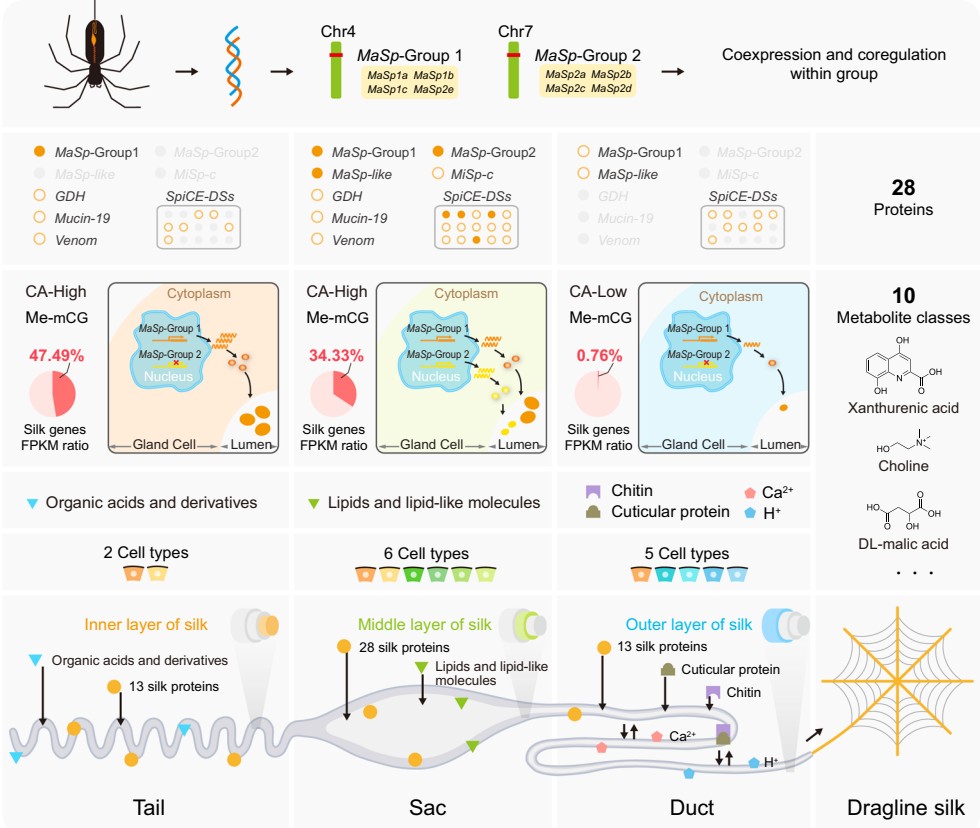

**Fig. 6 | Molecular basis of tri-sectional dragline silk generation in the *T. clavata* Ma gland.** Orange solid circles represent the genes with an FPKM > 10,000, and orange hollow circles represent the genes with an FPKM < 10,000, CA chromatin accessibility, Me methylation. Adobe Illustrator 2020 was used to create the image.

of these groups exhibited concerted SC and ST expression profiles in the tri-sectional Ma gland. We also identified the group-specific common TF motifs at the epigenetic level and constructed group-specific lncRNA-miRNA-mRNA networks at the ceRNA level. Such results revealed novel structural, expressional, and regulatory characteristics of spidroin genes that have not been reported in other spider genomes. Spidroins are thought to exert important influences on the mechanical properties of dragline silk[16,57,58]. As the group distributions of spidroin genes have already been identified in the high-quality spider genomes of *T. antipodiana*[35] and *Latrodectus elegans*[59] with cataloged spidroin proteins, our data fill an important information gap regarding the arrangement of spidroin genes on whole chromosomes and provide a general entry point for the mechanical differentiation of silk and the further study of spider genome evolution.

In addition to the above genomic findings, our results provide cellular clues for the tri-sectional organization and functional division of the Ma gland. While previous studies have shown that there are two or three cell types that secrete spidroins in the Ma gland of orb-web spiders[21,60], the number of cell types in the Ma gland has been a subject of debate. The divergent findings concerning these cell types have suggested that they may vary between species but probably also reflect technological difficulties in the capture of intact cells after sample preparation for cellular and morphological studies[20,21,61]. Our study provides well-verified scRNA-seq and ST data indicating the existence of ten cell types in the whole Ma gland, thus contributing to answering this contentious question. The fascinating single-cell spatial architecture of the gland further indicates the differentiation trajectory of Ma gland cells during cell state transitions, suggesting that the Sac and Duct are derived from the early cell type present in the Tail via cell differentiation.

Our genomic, proteomic, and metabolomic analyses further reveal details about the generation of silk in silk-producing glands. Physical and material studies have shown that liquid silk dope is transformed into insoluble fiber through the combined effects of pH and ion gradients as well as extensional and shear forces as it migrates through the silk gland[3,62]. Our work provides biological evidence of this role of the silk gland and further demonstrates high convergence of several molecular functions in the spider Ma gland Duct and the silkworm ASG, including calcium ion binding, chitin binding, signal transduction, and *V-type H⁺-transporting ATPase* expression. Such overlaps were silk gland-specific and not identified in other tissue types (hemocyte and ovary) (Supplementary Fig. 30). Furthermore, we identified convergent silk components in spider and silkworm silks, including two proteins (mucin-19 and GDH) and two major metabolites (choline and DL-malic acid). These analogous aspects indicated critical processes and components essential to silk formation, which contribute to understanding the evolution of silk spinning and can be used as references for the optimization of Duct-based artificial spinning devices and the design of silk protein solutions in artificial spinning.

In conclusion, our current work comprehensively illustrates the biological basis of dragline silk formation in an orb-weaving spider. To our knowledge, this study is the first to reveal the biological mechanism of silk spinning in spiders in such detail. We believe that the chromosome-scale reference genome of the golden orb-web spider and the molecular atlas of the tri-sectional Ma gland presented here will facilitate the understanding of the spider silk-spinning process in spiders and further serve as a powerful platform for evolutionary studies of silk-spinning organisms. More importantly, the significant molecular characteristics revealed by our results and the generated datasets should ultimately pave the way for producing optimized synthetic silks through genetic and biomimetic manipulations.

## Methods

### Karyotyping

*T. clavata* spiders were captured from the wild in Dali City, Yunnan Province, China. The captured adult female *T. clavata* spiders were cultured indoors at 25 °C, and their eggs were sampled on the third day after spawning for DNA karyotyping. Approximately 20 eggs were mashed with tweezers, mixed with 1.5 mL of 0.05% colchicine for three hours, immobilized with 0.075 mol/L KCl solution (20 min), and fixed in methanol:acetic acid (3:1) solution (30 min). The cell suspension was dropped onto precooled glass slides, flame-dried, stained with 5% Giemsa solution (50 min), rinsed with running water, and air-dried. Chromosome karyotypes were observed by using an Olympus microscope with 40× magnification.

### Short- and long-read transcriptome sequencing

**Short-read transcriptome sequencing.** To more comprehensively capture the gene set of *T. clavata*, the 18 following fresh tissues (Supplementary Data 1) were dissected in PBS solution: female body (TcF), male body (TcM), hemolymph (Hem), pedipalps (Ped), legs (Leg), epidermis (Epi), fat body (Fat), ovary (Ova), venom gland (Ven), whole major ampullate gland (Ma), tail of the Ma gland (Tail), sac of the Ma gland (Sac), duct of the Ma gland (Duct), minor ampullate gland (Mi), tubuliform gland (Tu), flagelliform gland (Fl), aggregate gland (Ag), and aciniform and pyriform glands (Ac & Py); and three independent biological replicates were performed for each sample. Then, total RNA was extracted using TRIzol reagent and sequenced on the DNBSEQ platform.

**Long-read transcriptome sequencing.** To improve the quality of spidroin gene annotation, we also performed full-length transcriptome sequencing. Seven silk gland tissues (Ma, Mi, Tu, Fl, Ag, Ac, and Py glands) were mixed to construct libraries for Isoform sequencing (Iso-seq) on the PacBio platform. To reduce fragment bias, we constructed two libraries: one ranging from 0–10 kb and the other from 5–10 kb.

### Genome assembly

For the de novo assembly of the *T. clavata* genome (Supplementary Fig. 1c), Nanopore long reads were initially corrected using Canu v2.2[63] with the default parameters, and SMARTdenovo (https://github.com/ruanjue/smartdenovo) was subsequently employed to produce the primary contigs. Then, three rounds of contig correction were performed using Racon v1.5.0[64]. To improve the accuracy of the assembly, Pilon v1.24[65] was utilized for contig polishing based on Illumina short reads. The paired Hi-C short reads were primarily trimmed by using Hic-pro v2.10[66], and the optimized reads were mapped to the draft contig with Juicer v1.6.2[67] using 3D-DNA v180419[68] for pseudochromosome construction and the Juicebox v1.9[67] tool for manual corrections.

### Genome annotation

**Repeat annotation.** A custom *T. clavata* repeat database was de novo identified by using RepeatModeler v2[69] and LTR_FINDER v1.06[70]. Then, the obtained repeat sequences were imported into RepeatMasker v4.05[71] to search for transposable elements (TEs) in the genome. Tandem Repeats Finder v4.09[72] was used to find tandem repeats.

**Gene annotation.** Gene structure prediction was based on homology data and de novo prediction. The data files included RNA-seq data (RNA-seq of 17 tissues), Iso-seq data, and protein sequences of closely related species (four species: *A. ventricosus*, *A. bruennichi*, *T. antipodiana*, and *T. clavipes*). AUGUSTUS v3.2.3[73] (http://bioinf.uni-greifswald.de/) software was used for de novo gene prediction. Evidence-based gene annotations were produced in the MAKER2[34] pipeline with the default parameters. Then, we integrated the MAKER and de novo results based on the following screening criteria: the proportion of repeat sequences was less than 50%, the protein sequence length was greater than 50 aa, the FPKM value was greater than 0.1 in at least ten samples, and the sequence was supported by Iso-seq evidence (identity > 95%) or a corresponding sequence from at least one closely related species (*E*-value > 1e⁻⁵, identity > 50%).

## Spidroin analysis

**Spidroin identification.** A total of 443 redundant spidroin sequences (Supplementary Data 6) were downloaded from the NCBI protein database (https://www.ncbi.nlm.nih.gov/protein). All protein sequences of *T. clavata* were used to align these spidroins by blastp (*E*-value < $1e^{-10}$), and 128 nonrepetitive hits were primarily identified. To further reliably identify spidroins, we double-checked that each sequence included typical repeats and N- or C-termini, and the final 28 putative spidroins were finally identified.

**Artificial full-length spidroin assembly.** Briefly, (1) 79 complete spidroin sequences, 48 N-terminal sequences, and 22 C-terminal sequences (Supplementary Data 7) were mapped to the *T. clavata* genome to generate the GFF track files using the genBlast v138[74] tool; (2) the BAM files of both the silk gland Iso-seq data and the silk gland RNA-seq data were extracted as expression data tracks; (3) the GFF3 files of the 28 spidroins were extracted as one track; (4) the track files from the previous three steps were imported into IGV v2.9.4[75], and the integrity of each spidroin was manually checked; and (5) the DNA sequence of the incomplete spidroin on the corresponding genome regions were truncated, then repaired based on frameshift translation and prediction using the online tool AUGUSTUS v3.2.3[73] (http://Augustus.gobics.de/).

**Spidroin motif analysis.** Amino acid content and motifs were quantified using a Perl script. The motifs included $(GA)_n$, $(A)_n$, GGX, XQQ, and GPGXX. O-glycosylation was predicted with the NetOGlyc v4.0[76] tool.

## Proteomics

The dragline silk of the spider *T. clavata* and cocoon silk of the silkworm *B. mori* were cut with scissors (Supplementary Data 1). First, silk samples ($n = 3$, 20 mg of each sample) were dissolved in 500 μL of 9 M LiSCN, and the supernatant was collected after centrifugation (10,000 × g, 10 min). Protein concentrations were measured via the Bradford protein assay (Beyotime, China). Then, the samples were diluted 10-fold in an 8 M urea solution, mixed with 5 × SDS–PAGE buffer, and separated using a NuPAGE 4–12% Bis-Tris protein gel (Thermo Fisher Scientific, USA). The samples were subsequently digested with trypsin (1/50 μg protein) for 20 h at 37 °C. The digested samples were lyophilized and resuspended in 0.1% formic acid (FA) and then measured via LC–MS/MS on a Q Exactive HF-X column (Thermo Fisher Scientific, USA). The gradient was set as 5–95% acetonitrile in 0.1% formic acid. The flow rate and analysis time were 600 nL/min and 70 min, respectively. The instrument parameters were as follows: full MS scan ranged from m/z 350 to 1500 with a resolution of 60,000 (at m/z 200); the automatic gain control (AGC) target value was $3 × 10^6$ and the maximum ion injection time was 20 ms; the top 40 precursors of the highest abundance in the full scan were selected and fragmented by higher energy collisional dissociation (HCD) and analyzed in MS/MS, where the resolution was 15,000 (at m/z 200), the AGC target value was $1 × 10^5$, the maximum ion injection time was 45 ms, a normalized collision energy was set as 27%, an intensity threshold was $2.2 × 10^4$, and the dynamic exclusion parameter was 20 s. The resulting raw MS data were analyzed with MaxQuant v1.3.0.5[77]. The MaxQuant searches were performed against the SpiderDB (https://spider.bioinfotoolkits.net) database containing 37,607 protein sequences. The search parameters were as follows: methionine oxidation and N-terminal acetylation were set as variable modifications, and carbamidomethyl cysteine was set as the fixed modification; the minimal peptide length was set to seven amino acids and a maximum of two miscleavages were allowed; both peptide and protein identifications were filtered at a 1% false discovery rate; the identified protein required at least one unique peptide and all common reverse hits and contaminants were removed. Based on the sum of peak intensities, the intensity-based absolute quantification (iBAQ) algorithm in MaxQuant was used to calculate the protein abundances.

## Metabolomics

Shredded dragline silk and cocoon silk samples ($n = 6$, 20 mg of each sample) were used for metabolite extraction (Supplementary Data 1). Each silk sample was placed in a tube, homogenized in 500 μL of 80% methanol, kept on ice for 10 min, and centrifuged at 15,000 × g for 20 min at 4 °C. The supernatant (250 μL per sample) was collected for LC–MS/MS experiments and injected onto a Hypesil Goldcolumn (C18, 100 × 2.1 mm, 1.9 μm) using a 17 min linear gradient at a flow rate of 0.2 mL/min. The eluents for the positive polarity mode were eluent A (0.1% FA in Water) and eluent B (Methanol). The eluents for the negative polarity mode were eluent A (5 mM ammonium acetate, pH 9.0) and eluent B (Methanol). The solvent gradient was set as follows: 2% B, 1.5 min; 2–100% B, 3 min; 100% B, 10 min; 100–2% B, 10.1 min; 2% B, 12 min. Q Exactive HF mass spectrometer (Thermo Fisher Scientific, USA) was operated in positive/negative polarity mode with a spray voltage of 3.5 kV, capillary temperature of 320 °C, sheath gas flow rate of 35 psi, and aux gas flow rate of 10 L/min, S-lens RF level of 60, Aux gas heater temperature of 350 °C. Full scan ranged from m/z 350 to 1500 with a resolution of 60,000 (at m/z 200). The AGC target value was $3 × 10^6$ and the maximum ion injection time was 20 ms. The top 40 precursors of the highest abundant in the full scan were selected and fragmented by HCD and analyzed in MS/MS, where the resolution was 45,000 (at m/z 200) for 10 plex, the AGC target value was $5 × 10^4$, the maximum ion injection time was 86 ms, a normalized collision energy was set as 32%, an intensity threshold was $1.2 × 10^5$, and the dynamic exclusion parameter was 20 s. Metabolite identities and quantities were confirmed by Compound Discoverer 3.1 (CD 3.1) software analysis, including alignment against the mzCloud (https://www.mzcloud.org/), mzVault (mzCloud Offline for mzVault 2.3_2020A.db), and Mass List (Endogenous Metabolite-Animal-POS/NEG-20191029) databases (retrieval date: 6/15/2020). Furthermore, the KEGG[78] (https://www.kegg.jp/), HMDB[79] (https://hmdb.ca/metabolites), and LIPIDMaps[80] (http://www.lipidmaps.org/) databases (retrieval date: 7/17/2020) were utilized for metabolite annotation.

## Methylation analysis

**Whole-genome bisulfite sequencing (WGBS).** The separated Tail, Sac, and Duct of the Ma gland were subjected to WGBS. First, genomic DNA was extracted and fragmented to a mean size of 250 bp by sonication using a Bioruptor (Diagenode, Belgium), followed by the addition of an "A" base to the 3' terminal ends of the DNA fragments and the ligation of methylated adaptors to both ends of DNA fragments. The EZ DNA Methylation-Gold Kit (ZYMO) was used for the bisulfite conversion of ligated DNA. Products were purified using a QIAquick Gel Extraction Kit (Qiagen, USA) and amplified by PCR. Finally, libraries were sequenced on the Illumina HiSeq 2500 platform.

**Calculation of methylation levels.** Clean reads were mapped to the reference genome by using BSMAP 2.90 software (https://code.google.com/archive/p/bsmap/) with the parameters (-v 8 -z 33 -p 4 -n 0 -w 20 -s 16 -f 10 -L 100) after removing the adaptors and low-quality reads ($N > 10\%$ and $Q < 20$). Methylation levels were estimated by determining the corresponding beta values: beta = M/(M + U), where M and U represent methylated and unmethylated signal values, respectively. The BED file of methylation data was converted to BigWig format by using the "bedGraphToBigWig" command line and visualized in IGV[75].

## Assay for transposase-accessible chromatin (ATAC)

**ATAC-seq library preparation and sequencing.** The separated Tail, Sac, and Duct of the Ma gland were subjected to ATAC sequencing. Samples were ground individually in liquid nitrogen and transferred to precooled cell lysate solution at 4 °C for 10 min. After centrifugation (500 × g, 4 °C, and 10 min), the supernatant was removed, and the cell pellets were retained. The pellets were washed with precooled buffer

solution and centrifuged (500 × g, 4 °C, and 10 min) to collect cell nuclei. The cell nucleus solution was mixed with Tn5-transposase and incubated for 30 min at 37 °C, followed by DNA fragment collection, PCR amplification, and sequencing on the Illumina PE150 platform.

**Chromatin accessibility evaluation.** Clean reads were mapped to the genome using Bowtie2[81] software, the data were converted from.sam format to.bam format, and the reads were filtered by using SAMtools v0.1.19[82] with the parameters (view -b -f 2 -q 30). The peaks of chromatin accessible regions were identified in each sample separately using MACS2[83] (−shift −100−extsize 200−nomodel -B -q 0.05). The "intersect" command of BEDTools v2.26.0[84] was used to identify shared peaks between replicate samples. Heatmaps were generated using the "plotHeatmap" function of deepTools v3.5.1[85]. Significantly enriched motifs were identified by the HOMER[86] tool.

## Whole-transcriptome analysis
Total RNA was extracted from the Tail, Sac, and Duct of the Ma gland with TRIzol reagent, followed by lncRNA and miRNA library construction, as described below.

**LncRNA sequencing and analysis.** Total RNA samples were incubated with DNase I to digest DNA fragments and with RNase H to remove rRNA. After RNA purification, cDNA was synthesized with an "A" base added to the 3′ end, and adapter ligation was performed, followed by UDG digestion of the second strand and PCR amplification. The constructed library was sequenced on the DNBSEQ platform. The clean reads were mapped to the reference genome using HISAT2[87] (−phred64−sensitive−no-discordant−no-mixed -I 1 -X 1000), and the transcripts were assembled using StringTie[88] (-f 0.3 -j 3 -c 5 -g 100 -s 10000 -p 8) and compared with known mRNAs using the CuffCompare (-p 12) program of Cufflinks[89]. Then, the cooperative data classification (CPC), txCdsPredict, and coding noncoding index (CNCI) methods were used to distinguish mRNAs and lncRNAs. Cis and trans methods were used to evaluate the target mRNAs of lncRNAs. Cis-lncRNAs were defined as those lncRNAs located 10 kb upstream or 20 kb downstream of the target mRNA. The binding energy (BE) of the lncRNAs and mRNAs was analyzed by RNAplex[90], and if the BE was < −30, the lncRNA was considered to be a trans-lncRNA.

**Small-RNA sequencing and analysis.** Total RNA was purified and isolated using PAGE electrophoresis and was then excised from the gel to obtain 18-30 nt RNA fragments. The 3′ end of each small RNA fragment was ligated to a 5-adenylated and 3-blocked adaptor, and a unique molecular identifiers (UMI)-labeled primer was added for hybridization with the 3′ adaptor. Then, the 5′ adaptor was hybridized with the 5′ end of the UMI label. The cDNA strand was synthesized with UMI-labeled primers, and highly sensitive polymerase was used to amplify the product. We used Bowtie2[81] (-q -L 16−phred64 -p 6) to align the clean reads to the reference genome and the Rfam small-RNA database[91] (using cmsearch with the parameters−cpu 6−noali). To identify miRNAs, which were annotated and predicted with the miRBase[92] database (https://www.mirbase.org/) and miRDeep2[93] tool, targets were predicted between miRNAs and lncRNAs/mRNAs using miRanda[43] (-en −20 -strict) and RNAhybrid[44] (-b 100 -c -f 2,8 -m 100000 -v 3 -u 3 -e −20 -p 1).

## Single-cell transcriptomics
**Sample processing and sequencing.** Freshly dissected Ma glands from three independent spiders were washed using culture medium and cut into small pieces, and tissue pieces were digested in a constant-temperature incubator for 45 min. After digestion, the medium was filtered through a 40 μm cell sieve and washed at least three times via centrifugation for 5 min at 300 × g and 4 °C. Finally, the cell suspensions with activity greater than 85% and agglomeration rates less than

5% were used for library construction and sequencing. The library was constructed according to the manufacturer's instructions (10 × Genomics) on the 10 × Chromium Single Cell 3′ Platform.

**Data analysis.** The raw data were imported into the cellranger v4.0.0 pipeline to obtain the expression matrices of each cell and each gene. To remove low-quality cells, cell numbers and gene UMI counts within the range of the median ± 2-fold the median absolute deviation (MAD) were retained for further analysis. Additionally, doublets (≥2 cells in one oil droplet) were removed using DoubletFinder v2.0.3[94]. A total of 9349 high-quality cells were used to analyze clustering and expression based on the Seurat v4.0.6[95] package. The developmental trajectories of all cell clusters were predicted with the Monocle v2.22.0[96] package.

## Spatial transcriptomics
**Sample processing and sequencing.** Fresh Ma glands were frozen with isopentane in liquid nitrogen, embedded with OCT, and stored in an airtight container at −80 °C. Embedded frozen tissue was sectioned, and sections with acceptable RNA quality (RIN > 7.0) were used for further experiments. Then, the sections were placed on Visium Spatial slides, fixed with methanol, stained with hematoxylin and eosin, and observed by a BX53 microscope (Olympus, Japan). The polyadenylated mRNA was captured with the primers employed for spots. RT Master Mix reverse transcription reagent was subsequently added to permeate tissue sections, and second-strand cDNA synthesis was performed on the slide by adding the Second Strand Mix. The obtained cDNA was denatured from each capture zone and transferred to the corresponding tube for amplification, library construction, and sequencing on the Illumina NovaSeq600 platform.

**Data analysis.** After removing adapters and low-quality (N > 3, Q < 5, and base ratio ≥ 20%) reads, the spaceranger v1.3.1 mkref and count programs were used for data preprocessing. Then, spot clustering was analyzed using the Seurat v4.0.6[95] package.

## Convergent evolution analysis
To evaluate the occurrence of convergent evolution between the silkworm silk gland and the spider Ma gland, we compared five aspects (see Supplementary note for more details): (1) the morphological structure of the silk glands; (2) the microstructures of dragline silk and cocoon silk observed under a SU3500 scanning electron microscopy (SEM) (Hitachi, Japan) with an accelerating voltage of 5 kV; (3) the expression convergence of silk gland genes, (4) the homologous protein components of dragline silk and cocoon silk; and (5) the metabolite components of dragline silk and cocoon silk. The orthologous genes shared by the silkworm and spider were identified using the blastp[97] program (E-value < 1e−5).

## Reporting summary
Further information on research design is available in the Nature Portfolio Reporting Summary linked to this article.

# Data availability
All high-throughput sequencing raw data in this project were deposited into the Genome Sequence Archive (GSA) of the National Genomics Data Center (NGDC, https://ngdc.cncb.ac.cn/) and are available through BioProject ID PRJCA014503. The mass spectrometry proteomics data have been deposited to the ProteomeXchange Consortium (http://proteomecentral.proteomexchange.org/cgi/GetDataset) via the iProX partner repository[98] with the dataset identifier PXD038734. The metabolomics data have been deposited to the NGDC (https://ngdc.cncb.ac.cn/omix/release/OMIX002862, https://ngdc.cncb.ac.cn/omix/release/OMIX002863). The single-cell and spatial metrices have been deposited in FigShare (https://figshare.com/articles/dataset/Single_cell_matrices_zip/20399475, https://figshare.com/articles/dataset/Spatial_

Transcriptomics_zip/20399580). The genome assembly, gene annotation, and multiomics analysis results of *Trichonephila clavata* are also available on SpiderDB (https://spider.bioinfotoolkits.net). Source data are provided for Figs. 1, 2, 4, 5. Source data are provided with this paper.

## Code availability

Custom scripts and workflows for data analysis are available at FigShare (https://figshare.com/articles/dataset/SpiderGenomeAnanlysis_code_zip/20399652, https://figshare.com/articles/dataset/SpiderGenomeAnanlysis_code1_zip/20399706).

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

## Acknowledgements

We thank Daojun Cheng at Southwest University and Yi Zou at Southwest University for critical reading of the manuscript. We thank Yazhou Li at Sichuan Agricultural University for providing help in karyotyping. We also thank BGI, Novogene, Frasergen, OE Biotech, and Berry Genomics companies for providing multiomics sequencing. This work was supported by the National Natural Science Foundation of China [U21A20248], National Natural Science Foundation of China [32000340], and Fundamental Research Funds for the Central Universities [XDJK2019TJ003].

## Author contributions

W.H., A.J., Q.X., and Y.W. conceptualized the project. Y.W. supervised the project. W.H., A.J., Z.Z., J.S., T.Y., T.X., Q.L., T.Y., J.Z., and Y.W. collected samples for multiomics sequencing. A.J. designed and implemented the genome assembly pipeline and performed data analysis. W.H. and A.J. drafted the manuscript. F.L., Y.L., Z.J., and Z.X. generated the data and built the database. W.H. and S.M. conducted the experiments. Y.W. finalized the manuscript with input from all authors. All authors read and approved the final manuscript.

## Competing interests

The authors declare no competing interests.
