## [Peer Review File · Nature Communications]

Reviewers' Comments:

Reviewer #1:

Remarks to the Author:

In this study, the authors performed a molecular atlas for the major ampullate (Ma) gland of the golden orb-web spider *Trichonephila clavata*. As described in the text, this atlas comprises a chromosome-scale reference genome and multiomics data, including RNA sequencing (RNA-seq), liquid chromatography–mass spectrometry (LC–MS), the assay for transposase-accessible chromatin with sequencing (ATAC-seq), bisulfite-seq, whole-transcriptomics (WT), single-cell (SC) RNA-seq, and spatial transcriptomics (ST) data. On the basis of these data, the authors revealed a hierarchical biosynthesis of spidroins, organic acids, lipids, and chitin in the sectionalized Ma gland. To probe the cellular characteristics of Ma gland, the authors performed SC RNA-seq analysis and identified ten cell types in the Ma gland. Generally, this study provided new data and insights into the Ma gland of *T. clavata*, which is of potential value. However, I had expected much more than what the manuscript told at this step.

This study attempt to generate a molecular atlas for the Ma gland of *T. clavata*, thereby elucidating the tri-sectional spinning mechanism of spider dragline silk. This could be an interesting study, however, there have been a lot of papers describing the genome and/or transcriptome of spider species and many interesting genes revealed based on these data, I did not find this study to make a more valuable contribution to the literature or to science.

First of all, it is obvious that the authors have read a number of literature related to the issues they discuss, but it is uncertain about the novelty of the scientific questions to be solved by this study. What is impressive in this study is that the authors analyzed the Ma gland of the *T. clavata* through chromosome-scale reference genome assembly and several sequencing technologies such as SC and ST RNA-seq. Of course, there is no doubt that all sequencing data have their own scientific value.

Furthermore, the whole manuscript is too much descriptive, with so many trivial findings in the text but without compelling highlights or emphasis, it is very hard to catch the novel point(s) from the current manuscript. In this study, both the chromosome-scale reference genome assembly and single-cell analysis on the Ma gland of *T. clavata* are actually interesting and valuable. The authors have attempted to describe the results of data analysis, including the expression of some representative genes, however, each part seems to be insufficient to reveal the key scientific questions. In depth analysis and necessary experimental verification are needed.

In summary, this work provided some new data and preliminary analysis on the Ma gland of *T. clavata*, it seems be more suitable for publication in the professional journals of data analysis.

Reviewer #2:

Remarks to the Author:

Hu et al. have put together a remarkably thorough investigation of the *Trichonephila clavata* genome and the major ampullate gland molecular biology. Quite impressive! Of particular note is the apparent high quality of the genome and the multi-layered approach to the gene expression and potential regulation in not only the whole major ampullate gland, but different sections and even determining individual cell types. Prior to publications, I'd like to see some clarifications and revisions.

Main concerns

1. Spidroins

a. Identification – I am worried about the inclusion of spidroins that lack a N and C-terminal domain, especially regarding ones with no previously annotated repeat structure or motifs. These would include Sp-AGL-rich, Sp-GP-rich. In general, the Sp... spidroins are not convincing based on motifs or expression patterns. The MaSp9 motif class reported in supplementary table S8 is not sufficient to convince me this is a spidroin. AgSp3 and AgSp4 motifs do look like AgSp1 and AgSp2 but Other proteins can generate similar sequences. The PySp motifs are not particularly convincing either. Significant BLASTp alignments of AgSp3, AgSp4 and PySp to the homologous spidroins in a search against NCBI nr might be more convincing.

b. Phylogenetic analysis – it looks like full-length sequences were used. The repetitive structure of spidroins makes interpreting phylogenetic reconstruction of full-length sequences extremely

challenging. In fact, the repeat sequences shouldn't even align across many types. I strongly recommend limiting phylogenetic analysis to the N and C-terminal domains. If this was done, please clarify.

c. Naming – by convention MaSp1 has represented spidroins with GGX and poly-A motifs and MaSp2 also included GPGXX. The sequential naming of the MaSp as 1-9 and then calling them group 1 and group 2 is confusing. I admit I'm also stumped as to the best way to name these, but if a MaSp has a motif unit typical of MaSp1, it should be called "MaSp1". Then to separate the multiple copies, they could be named "MaSp1 locus 1", "MaSp1 locus 2", etc. Similar for MaSp2. Some additional MaSp sequences have been identified so to be called MaSp3, etc the MaSp probably needs to have a best BLASTp alignment to one of those alternative spidroins. However, I admit that there is not a clear convention past MaSp1 and MaSp2.

d. Naming AgSp – Similar comment where AgSp1 has a very different repeat organization than AgSp2 as defined in Stellwagen & Renberg (G3, 2019).

2. Convergent evolution

a. While I found the comparisons with Bombyx silk glands interesting, I was not particularly convinced that there was much overlap in molecular functions other than the gross morphology of the major ampullate gland with the Bombyx gland.

b. My main concern is that I suspect if any two tissue types were compared across species, some overlap would be found, especially for tissues that are likely to be expressing a lot of extruded protein. Thus, I'd like to see comparisons between other tissues to exclude the possibility that there is simply going to be overlap if you dig deep enough.

Additional concerns, in order of encounter not importance.

3. Write out abbreviations on first use.

4. "Spiders are a group" should be reworded. "Spiders form a group" would be OK. Or "Spiders (Order Araneae) are abundant generalist..."

5. Reference numbering is not correct for many references. I am not sure where they go wrong but definitely the references in the second paragraph of the introduction cannot support the statements made in the paragraph. For instance, #22 and 23 are not for recombinant spidroin papers, and the SpiCE references are #34 and 35 in the bibliography but 24,25 are included in the text. There were many other instances throughout the manuscript where I couldn't find the reference that supported the statement made. These need to be updated throughout.

6. End of second paragraph of introduction – clarify which aspects of Ma gland biology are poorly understood. I would argue much is already known, although this manuscript certainly expands that knowledge base.

7. Line 57: change exhibited and constructed to present tense, since this species has not gone extinct.

8. Line 100: I think it is OK to use homologous or perhaps syntenic to refer to the similar grouping of MaSp sequences on chromosomes across species.

9. For any enrichment analysis, please clarify the comparison group.

a. For instance, line 168... GO terms significantly enriched in the tail... relative to all GO terms assigned to whole genome? Or something else?

b. As another instance, line 198... what is enrichment of TF motifs relative to? Are these just the TFs found in the ATAC-seq peak set, and then compared to all TF motifs? Or the ones in the 2kb upstream region that aren't in the ATAC-seq peak set?

10. Lines 184-188: I'd like a little more context for the mean RPKM values reported for chromatin accessibility. The means seem to swamp the higher signal of accessibility surrounding the transcription start sites (TSS) as shown in Figure 3a. Specifically, the peak of RPKM looks similar among the 3 regions of the MA gland, so the conclusion that the duct has low accessibility isn't entirely clear to me. Would describing the number of genes with greater than background RPKM for each section help? Or the extent of the accessibility? E.g. text that explains the pattern in Figure 3a might be more intuitive?

11. I have to admit my own ignorance that I had never seen the term ceRNA before. I think it would be useful to add a brief explanation of context to the results and first mention of ceRNA. E.g. are there really mRNAs that compete for microRNA regulation? What would this look like and why would it matter for dragline silk gene expression, regulation, silk formation? The methods for identifying the network in Figure 3g is not clear to me from the results, methods, or the supplementary note. Probably just some context language around the goals of miRanda and RNAhybrid would help here. These look like programs that identify miRNA binding sites. Was the

network, just build by hand after that? Or do these programs also implement network algorithms?

12. After teaching myself about ceRNA and figuring out what the network in Fig 3g is showing, I'm not convinced there is much opportunity for competition. I'm more convinced of the potential for coregulation of MaSp-1 and coregulation of MaSp-2. Again, context and clarification would help.
13. Line 209: Define WT as Whole Transcriptome.
14. Line 341: Prior work established importance of duct in protein structural transition. Please cite it here.
15. Line 358: "shared" should be "shear" (I think)
16. Line 370: What is SC and ST?
17. Line 371: discuss meaning of networks rather than relying on reader to intuit ceRNA
18. Lines 380-381: What is the debate regarding the Ma cell types? How does your work support one side or another, on no past work?
19. Lines 389-401: As mentioned above, I'm not convinced the analogous expression between spider MA and bombyx silk glands is that meaningful for silk production. Might want to remove this section, pull back on conclusions, or need to convince reader that other tissue types wouldn't have some overlap too.
20. Line 483: should "1,5000 x g" be "1,500 x g" or "15,000 x g"?
21. Line 501: there seems to be a word missing in "were by determining". Maybe "were estimated by determining"
22. Figure 4 legend: I think white-filled cycles and black-filled cycles should be circles instead of cycles. Also, what are the units of expression for part e?

-Nadia A. Ayoub

Reviewer #3:

Remarks to the Author:

This manuscript reports the (near) chromosomal level assembly of a golden orb-weaving spider *Trichonephila clavata* with the high-quality. The authors focus on molecular model analyses of spider silk gland (Ma silk gland). The comprehensive multiomics studies and applications are performed in this manuscript, showing their huge workload in terms of sampling and bioinformatics analyses. Such detailed biological mechanism of silk is an important foundation for future study about silk gland, like in evolution and silk spinning behavior. The presented work is quite extensive and the manuscript is well written and illustrated by clear informative figures.

A few other comments:

1) In major manuscript, authors use fragments per kilobase of transcript per million mapped reads (FPKM) to measure the expression abundance of genes (like line 179-181), but in figure 6, authors use RPKM, which confuses me. As far as I know, RPKM means reads per kilobase of exon model per million mapped reads. I would encourage authors to use same normalization method about RNA-Seq, or give an explanation in the legend.

2) CRISPR/Cas9-based experiment in the silkworm is executed, and the silk production (cocoon weight and cocoon layer rate) of the Hsp20-KO strain was significantly lower than those of the wild type. Does this imply that if the gene Hsp20 is knocked out in spiders would it have a similar effect?

3) In the part of convergent evolution of the tri-section silk gland between spider and silkworm, tail of spider silk gland and PSG of silk shows no sharing GO terms. Many GO terms are similar in secondary structure even though the GO numbers are different, and I'm worried that more similar information may be missed. In addition, whether there are same transcription factors can be checked in tail of spider silk gland and PSG of silk, based on the numerous orthologs between two species.

RESPONSE TO REVIEWER COMMENTS:

We thank the Reviewers for the time and effort spent carefully reviewing our manuscript and providing constructive comments. Point-by-point responses to all comments and modifications to the manuscript are listed below. The reviewers' comments are in plain text, and our responses are in blue. The cross-references to the manuscript are bold and underlined>. (Line numbers mentioned in the responses may not coincide with the original line numbers.)

Reviewer #1 (Remarks to the Author):

In this study, the authors performed a molecular atlas for the major ampullate (Ma) gland of the golden orb-web spider *Trichonephila clavata*. As described in the text, this atlas comprises a chromosome-scale reference genome and multiomics data, including RNA sequencing (RNA-seq), liquid chromatography–mass spectrometry (LC–MS), the assay for transposase-accessible chromatin with sequencing (ATAC-seq), bisulfite-seq, whole-transcriptomics (WT), single-cell (SC) RNA-seq, and spatial transcriptomics (ST) data. On the basis of these data, the authors revealed a hierarchical biosynthesis of spidroins, organic acids, lipids, and chitin in the sectionalized Ma gland. To probe the cellular characteristics of Ma gland, the authors performed SC RNA-seq analysis and identified ten cell types in the Ma gland. Generally, this study provided new data and insights into the Ma gland of *T. clavata*, which is of potential value. However, I had expected much more than what the manuscript told at this step.

This study attempt to generate a molecular atlas for the Ma gland of *T. clavata*, thereby elucidating the tri-sectional spinning mechanism of spider dragline silk. This could be an interesting study, however, there have been a lot of papers describing the genome and/or transcriptome of spider species and many interesting genes revealed based on these data, I did not find this study to make a more valuable contribution to the literature or to science.

First of all, it is obvious that the authors have read a number of literature related to the issues they discuss, but it is uncertain about the novelty of the scientific questions to be solved by this study. What is impressive in this study is that the authors analyzed the Ma gland of the *T. clavata* through chromosome-scale reference genome assembly and several sequencing technologies such as SC and ST RNA-seq. Of course, there is no doubt that all sequencing data have their own scientific value.

Furthermore, the whole manuscript is too much descriptive, with so many trivial findings in the text but without compelling highlights or emphasis, it is very hard to catch the novel point(s) from the current manuscript. In this study, both the chromosome-scale reference genome assembly and single-cell analysis on the Ma gland of *T. clavata* are actually interesting and valuable. The authors have attempted to describe the results of data analysis, including the expression of some representative genes, however, each part seems to be insufficient to reveal the key scientific questions. In depth analysis and necessary experimental verification are needed.

In summary, this work provided some new data and preliminary analysis on the Ma gland of *T. clavata*, it seems be more suitable for publication in the professional journals of data analysis.

Response: We thank the reviewer for the time and effort that you have put into reviewing the previous version of our manuscript. We appreciate the detailed and constructive comments. These comments are all valuable and very helpful for revising and improving our paper, as well as the important guiding significance to our research. We have studied the comments carefully and made corrections that we hope meet with approval.

The process of natural silk production in the spider major ampullate (Ma) gland endows dragline silk with extraordinary mechanical properties and the potential for biomimetic applications. Although many papers have described the genome and transcriptome of spider species and revealed many interesting genes based on these data¹⁻⁴, the precise genetic roles of the tri-sectional Ma gland during the silk-production process remain unknown. Most studies on the Ma gland tend to treat it as a whole rather than discuss it in the Tail, Sac, and Duct segments. Much of our current understanding of these segments is based on physical and material studies that have provided only a partial picture of their nature^{5,6}. For example, they are characterized by gradients of pH values, ion concentrations, and shear forces⁷⁻⁹. Therefore, it is necessary to divide the Ma gland into Tail, Sac, and Duct to explain the molecular biological mechanism of dragline silk formation. Our work performed the first molecular atlas of natural spider dragline silk production using genome assembly for the golden orb-web spider *Trichonephila clavata* and multiomics defining for the segmented major ampullate (Ma) gland: Tail, Sac, and Duct, which showed the precise gene regulation, protein secretion, cell architecture, and anatomic form of the Ma gland segments. The genetically mediated mechanisms of dragline silk production in the Ma gland will facilitate an understanding of the spider silk-spinning process and guide significance

for optimizing artificial spider silk and establishing a biomimetic system.

We agree with your valuable feedback that our research has so many trivial findings and the novelty of the scientific questions to be solved by this study is uncertain. As you said, some sections of this paper do have some shortcomings. Therefore, we have optimized our manuscript writing, conducted in-depth data analysis, made relevant corrections, and summarized the highlights of our study compared with previous spider genomic studies. This revised paper is tightly focused on the mechanism of dragline silk production and stands out in the following aspects:

(1). A high-quality spider genome is the cornerstone of the whole research. The *T. clavata* genome was assembled to the chromosome level by using the most comprehensive sequencing data (Supplementary Data 1). These data included 129 × Nanopore reads, 73 × Illumina reads, 161 × Hi-C reads, the transcriptome of 18 tissues (three replicates for each sample), and PacBio Iso-seq data (two libraries: 0–10 k and 5–10 k). Based on these data, we assembled a 2.72 Gb high-quality *T. clavata* genome with a heterozygosity of 1.44%. We found that DNA transposons and LTR retrotransposons were responsible for the enlargement of the *T. clavata* genome (Supplementary Fig. 7). Moreover, we also assembled highly repetitive spidroin sequences and found that MaSps and MiSps were clustered on chromosomes (Fig. 1c). The evolutionary analysis indicated that MaSp groups 1 and 2 have a recent common ancestral origin between species (Supplementary Fig. 12). These results provide a solid foundation for the paper and an important reference for subsequent spider-related studies.

(2). Multiomics analysis defines the proteomic and metabolic components of dragline silk and traces their origins from the tri-sectional Ma gland. Given that we are captivated by the incomparable performance of spider dragline silk, our goals are to comprehensively explore the substances and genes related to the synthesis of dragline silk and provide a reference for artificial spinning. Most studies have focused on some particular spidroins, which lack a comprehensive and in-depth study of dragline silk. This paper makes up for this shortcoming and elaborates the components of dragline silk and the silk production mechanism of the Ma gland in detail by using multiple omics data. The present study is the first to divide the spider Ma gland into three sections for multiomics analysis. Our multiomics analysis revealed the highlighted results:

(i). The MaSp content was the highest in dragline silk, and the organic acid content was the richest in the metabolites (Fig. 2b, c).

(ii). The synthesis of dragline silk was closely related to the organic acids of Tail, the lipids of Sac, and

the chitin of Duct (Fig. 2j).

(iii). The differential expression pattern of the MaSp groups was related to epigenetic modification and transcriptional regulation (Fig. 3).

(iv). The rich MaSp content in dragline silk was due to a large number of MaSp secretory cells in the Ma silk gland, and these cells were mainly concentrated in the Tail and Sac.

(v). We also characterized the spatial expression positions of genes related to organic acids, lipids, ions, and chitin in the silk glands, further explaining the rational and orderly synthesis of silk proteins in the silk gland lumen.

We believe that these characteristics can provide a new understanding of dragline silk production and are of guiding significance for the modification of dragline silks and artificial spider silk in vitro.

(3). The first single-cell spatial architecture at the whole-Ma-gland scale. We strongly agree that the single-cell (SC) RNA-seq and spatial transcriptomics (ST) data you mentioned are of scientific significance. For spiders, this is the first single-cell map of the spider's Ma gland. We used SC and ST data to identify cell types and spatial distribution and carried out various available analyses, including cell cluster classification, functional annotation, and pseudotime analysis. This greatly broadens the scope of spider research. Future researchers can take this map as a reference to carry out the precise transformation, development, and utilization of higher-value spider silk, etc., which is of great significance for the vertical and horizontal expansion of spider silk production. According to the single-cell expression profiles, we divided them into 10 cell clusters, which further verified the difference between the Tail, Sac, and Duct of the Ma gland. We found that the cell types with different signature genes were distributed in the Tail, Sac, and Duct regions, playing different functions to coordinate the production of dragline silk. Combined with ST data, it was further found that Sac and Duct were derived from the early cell types existing in the Tail through cell differentiation. Whether using multiomics data or single-cell RNA-seq data, we all found a difference between the Tail, Sac, and Duct of the Ma gland, which proved the necessity and significance of the analysis according to the Tail, Sac, and Duct structures.

(4). Molecular analogs between the spider Ma gland and silkworm silk gland for silk formation. We considered the silkworm silk gland as a bioreactor for the synthesis of spider silk. Therefore, we performed a multiomics comparison and analysis of silk and silk glands between silkworm and spider and found many convergent and similar substances and genes. However, due to the lack of functional

research on many spider genes and the lack of a stable genetic system, we cannot verify the function of genes through experiments, which is truly a pity. To overcome these difficulties, we verified the spinning mechanism of the Tail, Sac, and Duct in the Ma gland through the circuitous way of silkworms through genetic manipulation experiments and proved the possibility of the theory to a certain extent. The analogous molecular characteristics listed here further suggested another important referable method, silkworm-silk-gland-bioreactor, for spider silk synthesis.

(i). Both silkworms and spiders had similarities in the morphology and function of their silk glands. The posterior part mainly secretes silk proteins, the middle part is enriched in lipid-related terms, and the anterior part is enriched in ion- and chitin-related terms (Fig. 5a, b, and f).

(ii). There were a large number of genes with homologous and consistent expression trends in the silk gland tissues of silkworm and spider.

(iii). Silkworm CRISPR/Cas9 experiments also verified that spider homology genes were closely related to the yield of silk synthesis (Fig. 5d, e).

We also thank the Reviewer for the comments concerning our scientific writing. Indeed, we agree that the previous version of our manuscript was too descriptive, and it was difficult to capture the novel points. As advised, to focus on the mechanism of dragline silk production and stand out the above aspects, we have more accurately optimized the background, purpose, and significance of this study to the introduction section, summarized and condensed our results, added in-depth analyses to the results section, and added relevant notes to the discussion section of the manuscript. The revised portions are marked in blue in the revised paper.

In summary, our research is extensive and profound in the molecular biological mechanism of dragline silk production. The results of the *T. clavata* high-quality genome, Ma gland multiomics characteristics, and comparative analysis for silk glands between silkworm and spider expands the knowledge of spider silk production and may eventually benefit spider-inspired fiber innovations. In addition, we are concerned that the data value of this study will be overwhelmed and then developed the SpiderDB database (<https://spider.bioinfotoolkits.net>). The database was an open and accessible online resource for presenting multiomics data of *T. clavata* in an interactive user interface so that interested researchers could further study. Thank you again for your suggestions, and we hope to learn more from you.

Reviewer #2 (Remarks to the Author):

Hu et al. have put together a remarkably thorough investigation of the *Trichonephila clavata* genome and the major ampullate gland molecular biology. Quite impressive! Of particular note is the apparent high quality of the genome and the multi-layered approach to the gene expression and potential regulation in not only the whole major ampullate gland, but different sections and even determining individual cell types. Prior to publications, I'd like to see some clarifications and revisions.

Main concerns

1. Spidroins

a. Identification – I am worried about the inclusion of spidroins that lack a N and C-terminal domain, especially regarding ones with no previously annotated repeat structure or motifs. These would include Sp-AGL-rich, Sp-GP-rich. In general, the Sp... spidroins are not convincing based on motifs or expression patterns. The MaSp9 motif class reported in supplementary table S8 is not sufficient to convince me this is a spidroin. AgSp3 and AgSp4 motifs do look like AgSp1 and AgSp2 but Other proteins can generate similar sequences. The PySp motifs are not particularly convincing either. Significant BLASTp alignments of AgSp3, AgSp4 and PySp to the homologous spidroins in a search against NCBI nr might be more convincing.

Response: We thank the Reviewer for the comments and agree with the suggestions. We apologize for our inaccurate description of spidroins that lack N- and C-terminal domains. In fact, the “spidroins that lack N- and C-terminal domains” indicate that the spidroins lack typical/conserved N- and C-terminal domains. Among the 28 spidroins we identified, 26 have nonrepetitive N- and C-terminal domains, and 15 have typical/conserved N- and C-terminal domains. We have corrected Supplementary Fig. 11 (Figure R1) and rewritten the relevant descriptive sentence in the manuscript.

Figure R1: Protein structure of 28 spidroins. Green and orange boxes indicate the typical/conserved N- and C-terminal domains of spidroins. Gray boxes indicate the atypical N- and C-terminal domains.

As advised, we downloaded 433 redundant spidroin sequences from the NCBI protein database (<https://www.ncbi.nlm.nih.gov/protein>). The protein sequences of AgSp3, AgSp4, PySp, Sp-AGL-rich, and Sp-GP-rich were used to align these spidroins by blastp (E-value < $1e^{-5}$). As shown in Figure R2a-d, we generated network diagrams of AgSp3 (renamed AgSp3a), AgSp4 (renamed AgSp2), MaSp9 (renamed MaSp-like), Sp-AGL-rich, Sp-GP-rich, and PySp and found that all of them were aligned to the reference spidroin sequences, suggesting that they are spidroin-like proteins.

MaSp9 (renamed MaSp-like) has nonrepetitive N&C-terminal domains and a repetitive domain (Figure R2e). The sequences of GGX, $(GA)_n$, and $(A)_n$ are typical motifs in MaSp^{10,11}. We found that the repetitive domain of MaSp9 lacked the $(A)_n$ motif but was rich in GGX and $(GA)_n$ motifs. Therefore, we redefined the MaSp9 protein as a MaSp-like protein.

Figure R2: **a-d**, Network diagram of AgSp2 (a), AgSp3a (a), PySp (b), Sp-AGL-rich (c), and Sp-GP-rich (d). Red circles indicate the query protein, and other circles indicate reference spidroins. **e**, Protein sequence of MaSp-like showing GGX- and (GA)_n-rich motifs.

b. Phylogenetic analysis – it looks like full-length sequences were used. The repetitive structure of spidroins makes interpreting phylogenetic reconstruction of full-length sequences extremely challenging. In fact, the repeat sequences shouldn't even align across many types. I strongly recommend limiting phylogenetic analysis to the N and C-terminal domains. If this was done, please clarify.

Response: We thank the Reviewer for the comments and agree with the suggestions. In the manuscript, we used the full-length sequences to construct the phylogenetic tree due to the lack of a nonrepetitive

N/C-terminal domain in two spidroins (AgSp2 and AgSp3b). However, it was indeed challenging. As you suggested, we reconstructed the phylogenetic tree of spidroins using the N-terminal domain because the N-terminal domain has a longer sequence than the C-terminal domain. We have corrected Supplementary Fig. 12a and renamed all spidroins depending on this phylogenetic tree (Figure R3).

Figure R3: Phylogenetic analysis of 28 spidroins. The BmFibH of *Bombyx mori* is an out-group.

c. Naming – by convention MaSp1 has represented spidroins with GGX and poly-A motifs, and MaSp2 also included GPGXX. The sequential naming of the MaSp as 1-9 and then calling them group 1 and group 2 is confusing. I admit I’m also stumped as to the best way to name these, but if a MaSp has a motif unit typical of MaSp1, it should be called “MaSp1”. Then to separate the multiple copies, they could be named “MaSp1 locus 1”, “MaSp1 locus 2”, etc. Similar for MaSp2. Some additional MaSp sequences have been identified so to be called MaSp3, etc the MaSp probably needs to have a best BLASTp alignment to one of those alternative spidroins. However, I admit that there is not a clear convention past MaSp1 and MaSp2.

Response: We thank the Reviewer for the comments and agree with the suggestions. Indeed, we are stumped by the spidroin naming. According to the suggestions, we have renamed the MaSpS that had GGX and (A)_n motifs “MaSp1” and that had GPGXX, GGX, and (A)_n motifs “MaSp2”. The subclassification

of MaSp1s (MaSp1a–c) and MaSp2s (MaSp2a–e) was performed by sequence alignment and phylogenetic analysis using the N terminal domain (Figure R4).

Figure R4: **a**, Motif analysis of MaSp. Yellow boxes indicate proteins containing the corresponding motif. **b**, Phylogenetic analysis of MaSp.

d. Naming AgSp – Similar comment where AgSp1 has a very different repeat organization than AgSp2 as defined in Stellwagen & Renberg (G3, 2019).

Response: We thank the Reviewer for the suggestion. We downloaded AgSp1 and AgSp2 sequences of *A. trifasciata* reported by Stellwagen & Renberg (G3, 2019)¹². According to the sequence alignment and phylogenetic analysis of AgSp between *T. clavata* and *A. trifasciata*, we renamed the AgSp of *T. clavata* (Figure R5).

Figure R5: **a-b**, Sequence alignment (a) and phylogenetic analysis (b) of AgSp between *T. clavata* and *A. trifasciata*.

2. Convergent evolution

a. While I found the comparisons with Bombyx silk glands interesting, I was not particularly convinced that there was much overlap in molecular functions other than the gross morphology of the major ampullate gland with the Bombyx gland.

Response: We thank the Reviewer for the comment. As the Reviewer said, the major ampullate gland

and the *Bombyx* gland were grossly similar in morphology. We proposed molecular functional similarities between the major ampullate gland and the *Bombyx* gland based on the following points:

(1). Previous studies have found pH gradients in both the major ampullate gland and the *Bombyx* gland, which from base to acid form Tail (7.6)/PSG (8.2) to Duct (5.7)/ASG (6.2)^{7,8,13}. Carbonic anhydrase was found to be responsible for the generation and maintenance of the pH gradient of both the major ampullate gland and the *Bombyx* gland^{7,8,13}. The pH gradients have been inferred to be important for fiber formation⁸.

(2). Previous studies have found ion concentration gradients in both the major ampullate gland and the *Bombyx* gland. For example, the sodium levels were found to increase from the distal to proximal parts of both the major ampullate gland and *Bombyx* gland using X-ray emission^{9,14}. An increased concentration of sodium chloride is important in destabilizing the silk protein homodimer¹⁵.

(3). Previous studies have found chitin in the cuticle of the spinning ducts of both the major ampullate gland and the *Bombyx* gland^{16,17}. Chitin is an essential component for the construction of spinning ducts and for generating shear forces^{8,16,17}

(4). Our results found similarities between the major ampullate gland and the *Bombyx* gland, including calcium ion binding, chitin binding, signal transduction, and *V-type H⁺-transporting ATPase* expression. Furthermore, we identified convergent silk components in spider and silkworm silks, including two proteins (mucin-19 and GDH) and two major metabolites (choline and DL-malic acid). Some of the analogous aspects have been reported previously, and some of them were novelty identified, which needed to be further studied.

In summary, we propose that there are molecular functional similarities between the major ampullate gland and the *Bombyx* gland in the silk formation processes of the silk gland.

b. My main concern is that I suspect if any two tissue types were compared across species, some overlap would be found, especially for tissues that are likely to be expressing a lot of extruded protein. Thus, I'd like to see comparisons between other tissues to exclude the possibility that there is simply going to be overlap if you dig deep enough.

Response: We thank the Reviewer for the comment. As advised, we have added two other *T. clavata* tissues (hemocyte and ovary) to the Venn comparison analysis. We identified 5,319 and 2,222 tissue-

specific expressed genes in hemocyte and ovary, respectively (Figure R6a-b). Gene Ontology (GO) analysis identified 354 and 67 GO terms enriched in hemocyte and ovary, respectively (Figure R6c-d). The shared GO terms of different tissues indicated the analogous aspects in tissue-developing and function-performing. As shown in Figure R6c, no GO term was identified in the comparison groups of Hemocyte vs. Tail, Hemocyte vs. Sac, Hemocyte vs. PSG, and Hemocyte vs. ASG. Only three shared GO terms (cytoskeleton organization, cytoskeletal protein binding, and GTPase activity) and one shared GO term (NAD binding) were identified in the hemocyte vs. Duct and hemocyte vs. MSG comparison groups, respectively. As shown in Figure R6d, no GO term was identified in the comparison groups of Ovary vs. Tail, Ovary vs. Sac, Ovary vs. PSG, Ovary vs. MSG, and Ovary vs. ASG. Only three shared GO terms (microtubule-based process, cytoskeleton organization, and supramolecular complex) were identified in the comparison of Ovary vs. Duct. Our results showed that the overlap analysis in different comparison groups exhibited differentiated GO terms. The shared GO terms between spider and silkworm silk glands were specific and not identified when we compared silk glands with other tissue types (hemocyte and ovary).

Figure R6: **a-b**, Tissue-specific gene expression analysis of the *T. clavata* hemocyte (a) and ovary (b). **c-d**, GO enrichment and Venn comparison analyses of hemocyte vs. silk gland (c) and ovary vs. silk gland (d). A *P* value < 0.05 was set as the criterion for screening enriched GO terms.

Additional concerns, in order of encounter not importance.

3. Write out abbreviations on first use.

Response: We thank the Reviewer for the suggestion. We have checked the full manuscript and written

out abbreviations on first use.

4. "Spiders are a group" should be reworded. "Spiders form a group" would be OK. Or "Spiders (Order Araneae) are abundant generalist..."

Response: We thank the Reviewer for the suggestion. We have reworded this sentence to "**Spiders (Order Araneae) are abundant generalist arthropod predators, including more than fifty thousand extant species.**"

5. Reference numbering is not correct for many references. I am not sure where they go wrong but definitely the references in the second paragraph of the introduction cannot support the statements made in the paragraph. For instance, #22 and 23 are not for recombinant spider papers, and the SpiCE references are #34 and 35 in the bibliography but 24,25 are included in the text. There were many other instances throughout the manuscript where I couldn't find the reference that supported the statement made. These need to be updated throughout.

Response: We thank the Reviewer for the suggestion. We have corrected the incorrect reference number, added appropriate reference citations to support the statement made, and updated all references throughout the manuscript.

6. End of second paragraph of introduction – clarify which aspects of Ma gland biology are poorly understood. I would argue much is already known, although this manuscript certainly expands that knowledge base.

Response: We thank the Reviewer for the suggestion. We agree with the Reviewer's argument and have rewritten the sentence to "**It is becoming apparent that the detailed mechanisms underlying dragline silk production, including the cellular architecture and molecular function of the Ma gland as well as the biocomposition and formation of dragline silk, are fundamental to advanced fiber innovation.**"

7. Line 57: change exhibited and constructed to present tense, since this species has not gone extinct.

Response: We thank the Reviewer for the suggestion. We have corrected the verb “exhibited” to “exhibits”.

8. Line 100: I think it is OK to use homologous or perhaps syntenic to refer to the similar grouping of MaSp sequences on chromosomes across species.

Response: We thank the Reviewer for the suggestion. We have corrected the adjective “analogous” to “homologous” and used “homologous” to refer to the similar grouping of MaSp sequences on chromosomes across species.

9. For any enrichment analysis, please clarify the comparison group.

Response: We thank the Reviewer for the suggestion. We have clarified the comparison group for each enrichment analysis.

a. For instance, line 168... GO terms significantly enriched in the tail... relative to all GO terms assigned to whole genome? Or something else?

Response: The GO terms significantly enriched in the Tail were relative to the GO terms enriched in the other two segments of the Ma gland, the Duct and Sac. We have rewritten the sentence to “**We found that the GO terms that were significantly enriched in the Tail (relative to the Duct and Sac) were mainly related to the synthesis of organic acids (the largest group in the silk metabolome), those in the Sac (relative to the Duct and Tail) were mainly related to the synthesis of lipids (the third-largest group in the silk metabolome), and those in the Duct (relative to the Sac and Tail) were related to ion (Ca²⁺ and H⁺) exchange and chitin synthesis (Fig. 2j).**”

b. As another instance, line 198... what is enrichment of TF motifs relative to? Are these just the TFs found in the ATAC-seq peak set, and then compared to all TF motifs? Or the ones in the 2kb upstream

region that aren't in the ATAC-seq peak set?

Response: The “enriched TF motifs” in the manuscript were predicted TF binding sites with E-values $< 1e^{-10}$ when compared the target peaks regions with the genome-wide peaks regions for each tissue. For example, we want to obtain all enriched TF motifs in the upstream and downstream 2k ranges of genome genes in Duct tissue, and we compared enriched TF motifs in these chromatin open regions with enriched TF motifs in the chromatin open regions of the entire genome. We apologize for our confusing expression. We have corrected “enriched TF motifs” to “potential TF motifs” and rewritten the sentence to **“We analyzed potential TF motifs among the ATAC-seq peak sets in the 2 kb regions upstream of the transcriptional start sites (TSSs).”** The “enriched TF motifs” in the following sentences and Figure 3 legend have also been corrected.

10. Lines 184-188: I'd like a little more context for the mean RPKM values reported for chromatin accessibility. The means seem to swamp the higher signal of accessibility surrounding the transcription start sites (TSS) as shown in Figure 3a. Specifically, the peak of RPKM looks similar among the 3 regions of the MA gland, so the conclusion that the duct has low accessibility isn't entirely clear to me. Would describing the number of genes with greater than background RPKM for each section help? Or the extent of the accessibility? E.g. text that explains the pattern in Figure 3a might be more intuitive?

Response: We thank the Reviewer for the suggestion. As advised, we added two sets of data to illustrate the chromatin accessibility (CA) of each section. We counted the number of significant ATAC peaks with greater than background RPKM for each silk gland segment. **“A total of 702,037 (Tail), 767,517 (Sac), and 653,361 (Duct) significant ATAC peaks (RPKM > 2) were identified in the 2 kb regions upstream and downstream of genes, and 10,501,151 (Tail), 11,356,55 (Sac), and 9,778,368 (Duct) significant ATAC peaks (RPKM > 2) were identified at the whole-genome level.”** These results suggested that the duct has relatively low accessibility compared with the Tail and Sac. We have added an explanation of Figure 3a in the legend and added the comparison of the number of significant ATAC peaks in the manuscript.

11. I have to admit my own ignorance that I had never seen the term ceRNA before. I think it would be

useful to add a brief explanation of context to the results and first mention of ceRNA. E.g. are there really mRNAs that compete for microRNA regulation? What would this look like and why would it matter for dragline silk gene expression, regulation, silk formation? The methods for identifying the network in Figure 3g is not clear to me from the results, methods, or the supplementary note. Probably just some context language around the goals of miRanda and RNAhybrid would help here. These look like programs that identify miRNA binding sites. Was the network, just build by hand after that? Or do these programs also implement network algorithms?

Response: We thank the Reviewer for the suggestion. Based on the suggestions, we have rewritten the sentences in the ceRNA section as follows: 1) We added a brief explanation of the context of the first mention of ceRNA in the manuscript: **“To investigate the impact of competing endogenous RNAs (ceRNAs: a post-transcriptional regulatory system implemented by miRNA and lncRNA¹⁸) corresponding to the regulation of dragline silk genes, we performed a whole-transcriptomic analysis of the tri-sectional_Ma gland...”**. 2) We added the expression pattern of the lncRNA and miRNA to explain the potential regulation pattern of the lncRNA–miRNA–mRNA competition network for dragline silk gene expression: **“Remarkably, we noted that the ceRNA networks of *MaSp1a–c* & *MaSp2e* (*MaSp*-Group 1) were tightly clustered, as were those of *MaSp2a–d* (*MaSp*-Group 2); three lncRNAs (LXLOC_047988, LXLOC_047990, and LXLOC_051464) in the *MaSp*-Group 1 network were highly expressed in the Ma gland (Supplementary Fig. 18); one lncRNA (LXLOC_070389) and four miRNAs (novel mir42, novel mir46, novel mir166, and miR-285 3) in the *MaSp*-Group 2 network were highly expressed in the Ma gland (Supplementary Fig. 18); in addition, the ceRNA networks of the two *MaSp* groups were independent of each other (Fig. 3g; Supplementary Fig. 18). These results further revealed potential post-transcriptional networks and the differentiated coregulatory pattern of the genes in the two *MaSp* groups in the Ma gland”**. 3) We added some context language around the goals of miRanda and RNAhybrid: **“From these data, we constructed potential lncRNA–miRNA–mRNA interaction pairs by using the miRanda¹⁹ and RNAhybrid²⁰ algorithms to identify the potential binding site between miRNA and lncRNA/mRNA, and then”**. 4) We added an explanation of the software used to construct and visualize networks: **“and then visualized the interaction networks by using Cytoscape software^{21”}**.

12. After teaching myself about ceRNA and figuring out what the network in Fig 3g is showing, I'm not convinced there is much opportunity for competition. I'm more convinced of the potential for coregulation of MaSp-1 and coregulation of MaSp-2. Again, context and clarification would help.

Response: We thank the Reviewer for the suggestion. We found that the mRNAs of *MaSp1a-c* & *MaSp2e* (*MaSp*-Group 1) tightly interacted with seven lncRNAs and two miRNAs, and the mRNA of *MaSp2a-d* (*MaSp*-Group 2) tightly interacted with four lncRNAs and four miRNAs (Fig. 3g). Among them, three lncRNAs (LXLOC_047988, LXLOC_047990, and LXLOC_051464) in the *MaSp*-Group 1 network were highly expressed in the Ma gland (Supplementary Fig. 18); one lncRNA (LXLOC_070389) and four miRNAs (novel_mir42, novel_mir46, novel_mir166, and miR-285_3) in the *MaSp*-Group 2 network were highly expressed in the Ma gland (Supplementary Fig. 18). Therefore, we suspected a potential regulatory pattern of the lncRNA-miRNA-mRNA competition network for *MaSp*-Group 1 and *MaSp*-Group 2 gene expression. However, their regulation patterns were independent of each other. As advised, we have added the context and clarification to the coregulation of *MaSp*-Group 1 and the coregulation of *MaSp*-Group 2: **"These results further revealed potential post-transcriptional networks and the differentiated coregulatory pattern of the genes in the two *MaSp* groups in the Ma gland"**.

13. Line 209: Define WT as Whole Transcriptome.

Response: We thank the Reviewer for the suggestion. To avoid the confusing use of abbreviations between whole-transcriptome and wild type, we have defined WT as wild type and removed all of the abbreviations for whole-transcriptome in the manuscript.

14. Line 341: Prior work established importance of duct in protein structural transition. Please cite it here.

Response: We thank the Reviewer for the suggestion. We have cited relevant references on the importance of Duct in protein structural transition.

15. Line 358: “shared” should be “shear” (I think)

Response: We thank the Reviewer for the suggestion. We have corrected “shared forces” to “shear forces”.

16. Line 370: What is SC and ST?

Response: We have defined abbreviations SC as single-cell and ST as spatial transcriptomics in the manuscript.

17. Line 371: discuss meaning of networks rather than relying on reader to intuit ceRNA

Response: We thank the Reviewer for the suggestion. We have added the relevant discussion to the meaning of ceRNA networks and their potential regulation pattern: “Our results provide genomic clues for the hierarchically ordered biosynthesis of spidroins. We documented that the *MaSp1a-c* & *MaSp2e*, *MaSp2a-d*, and *MiSp-a-e* genes are distributed in three distinct groups. In addition, we demonstrated that the *MaSps* within each of these groups exhibited concerted SC and ST expression profiles in the tri-sectional Ma gland. We also identified the group-specific common TF motifs at the epigenetic level and constructed group-specific lncRNA-miRNA-mRNA networks at the ceRNA level. Such results revealed novel structural, expressional, and regulatory characteristics of spidroin genes that have not been reported in other spider genomes.”

18. Lines 380-381: What is the debate regarding the Ma cell types? How does your work support one side or another, on no past work?

Response: The number of cell types in the Ma gland has been a subject of debate. Based on tissue sectioning, staining, and optical observations, previous studies have shown that there are two or three cell types in the Ma gland of orb-web spiders^{8,22}. However, our study provides single-cell sequencing data indicating the existence of ten cell types in the whole Ma gland, which supports no past work.

19. Lines 389-401: As mentioned above, I'm not convinced the analogous expression between spider MA and bombyx silk glands is that meaningful for silk production. Might want to remove this section, pull back on conclusions, or need to convince reader that other tissue types wouldn't have some overlap too.

Response: We thank the Reviewer for the suggestion. We have performed two *T. clavata* tissues (hemocyte and ovary) in the Venn comparison analysis. As shown in Figure R6a-b, no GO term was identified in the comparison groups of Hemocyte vs. Tail, Hemocyte vs. Sac, Hemocyte vs. PSG, Hemocyte vs. ASG, Ovary vs. Tail, Ovary vs. Sac, Ovary vs. PSG, Ovary vs. MSG, and Ovary vs. ASG. Three, one, and three shared GO terms were identified in the hemocyte vs. Duct, hemocyte vs. MSG, and Ovary vs. Duct comparison groups, respectively. More importantly, the overlapping GO terms in different comparison groups were differentiated. The shared GO terms indicated the analogous aspects in the tissue developing and performing functions. In summary, the nonsilk gland tissue types did not have any overlap, which was consistent with the comparison group of spider Ma vs. *Bombyx* silk glands. We have added necessary notes to the discussion section of the manuscript: **“Our work provides biological evidence of this role of the silk gland and further demonstrates high convergence of several molecular functions in the spider Ma gland Duct and the silkworm ASG, including calcium ion binding, chitin binding, signal transduction, and V-type H⁺-transporting ATPase expression. Such overlaps were silk gland-specific and not identified in other tissue types (hemocyte and ovary) (Supplementary Fig. 30).”**

20. Line 483: should “1,5000 x g” be “1,500 x g” or “15,000 x g”?

Response: We thank the Reviewer for the suggestion. We have corrected “1,5000 x g” to **“15,000 x g”**.

21. Line 501: there seems to be a word missing in “were by determining”. Maybe “were estimated by determining”

Response: We thank the Reviewer for the suggestion. We have added the missing word **“estimated”**.

22. Figure 4 legend: I think white-filled cycles and black-filled cycles should be circles instead of cycles. Also, what are the units of expression for part e?

Response: We thank the Reviewer for the suggestion. We have corrected “white-filled cycles” and “black-filled cycles” to “white-filled circles” and “black-filled circles”, respectively. Additionally, we have added the unit of expression, $\text{Log}_2(\text{transcripts per kilobase million; TPM})$, for part e in Figure 4e.

Reviewer #3 (Remarks to the Author):

This manuscript reports the (near) chromosomal level assembly of a golden orb-weaving spider *Trichonephila clavata* with the high-quality. The authors focus on molecular model analyses of spider silk gland (Ma silk gland). The comprehensive multiomics studies and applications are performed in this manuscript, showing their huge workload in terms of sampling and bioinformatics analyses. Such detailed biological mechanism of silk is an important foundation for future study about silk gland, like in evolution and silk spinning behavior. The presented work is quite extensive and the manuscript is well written and illustrated by clear informative figures.

A few other comments:

1) In major manuscript, authors use fragments per kilobase of transcript per million mapped reads (FPKM) to measure the expression abundance of genes (like line 179-181), but in figure 6, authors use RPKM, which confuses me. As far as I know, RPKM means reads per kilobase of exon model per million mapped reads. I would encourage authors to use same normalization method about RNA-Seq, or give an explanation in the legend.

Response: We thank the Reviewer for the suggestion. We apologize for this mistake, the misuse of RPKM in Figure 6 legend. The FPKM was used to measure the expression abundance of genes. We have corrected “RPKM” to “FPKM” in Figure 6 legend.

2) CRISPR/Cas9-based experiment in the silkworm is executed, and the silk production (cocoon weight

and cocoon layer rate) of the Hsp20-KO strain was significantly lower than those of the wild type. Does this imply that if the gene Hsp20 is knocked out in spiders would it have a similar effect?

Response: We agree with the Reviewer's comments. In cells, the accumulation of damaged proteins, resulting from mutation/misfolding/aggregation, can perturb cellular homeostasis and endanger survival under severe stress conditions^{23,24}. The *Hsp20* gene encodes a small heat shock protein that acts as a protein chaperone to protect other proteins against misfolding and aggregation²³. Based on the following reasons: 1) the silk gland is an organ assembling and secreting large amounts of protein and requires the protein damage protector, 2) the pair of *Hsp20* genes was orthologous with a high sequence identity of 87.1% between *T. clavata* and *B. mori*, 3) the pair of *Hsp20* genes was expressed in the silk glands of both *T. clavata* and *B. mori*, 4) the *Hsp20* gene served as a marker gene of SC cluster 5 of the *T. clavata* Ma gland, and 5) the silkworm *Hsp20*-KO strain presented a decreased silk production ability, we suspect that knocking out *the Hsp20* gene in spiders would cause phenotypes similar to those in silkworms (decreasing silk production).

3) In the part of convergent evolution of the tri-section silk gland between spider and silkworm, tail of spider silk gland and PSG of silk shows no sharing GO terms. Many GO terms are similar in secondary structure even though the GO numbers are different, and I'm worried that more similar information may be missed. In addition, whether there are same transcription factors can be checked in tail of spider silk gland and PSG of silk, based on the numerous orthologs between two species.

Response: We thank the Reviewer for the suggestion. As advised, we have investigated the shared GO term at level 2. As shown in Figure R7a, there were ten shared level 2 GO terms between the silk glands of spider and silkworm, including 1) transporter activity, biological regulation, and binding in the comparison group of Duct vs. ASG; 2) metabolic process, cellular process, catalytic activity, cellular anatomical entity, and binding in the comparison group of Sac vs. MSG; and 3) catalytic activity and metabolic process in the comparison group of Tail vs. PSG.

We also screened the same transcription factors (TFs) and found 285, 257, and 241 expressed orthologous TFs in the Duct & ASG, Sac & MSG, and Tail & PSG groups, respectively (Figure R7b). The top 10 TF types are listed in Figure R7c. We found that three ortholog TF types (zf-C2H2, bZIP, and zf-

NF-X1) were significantly upregulated in both the Duct and ASG (Figure R7d), and none of the ortholog TF types was significantly upregulated in Sac & MSG and Tail & PSG.

Figure R7: **a**, Shared GO terms in the comparison groups of Duct vs. ASG, Sac vs. MSG, and Tail vs. PSG. A P value < 0.05 was set as the criterion for screening enriched GO terms. **b**, Screening of the expressed orthologous TFs in the Duct & ASG, Sac & MSG, and Tail & PSG. **c**, Top 10 TF types in the Duct & ASG, Sac & MSG, and Tail & PSG. **d**, TFs expression convergence between the silk glands of *T. clavata* and *B. mori*.

References

- Babb, P.L. *et al.* The *Nephila clavipes* genome highlights the diversity of spider silk genes and their complex expression. *Nat. Genet.* **49**, 895-903 (2017).
- Kono, N. *et al.* Darwin's bark spider shares a spidroin repertoire with *Caerostris extrusa* but achieves extraordinary silk toughness through gene expression. *Open Biol.* **11**, 210242 (2021).
- Kono, N. *et al.* Orb-weaving spider *Araneus ventricosus* genome elucidates the spidroin gene catalogue. *Sci. Rep.* **9**, 8380 (2019).
- Kono, N. *et al.* Multicomponent nature underlies the extraordinary mechanical properties of spider dragline silk. *Proc. Natl. Acad. Sci. USA* **118**(2021).
- Li, J. *et al.* Spider silk-inspired artificial fibers. *Adv. Sci.* **9**, e2103965 (2022).
- Blamires, S.J., Blackledge, T.A. & Tso, I.M. Physicochemical property variation in spider silk: ecology, evolution, and synthetic production. *Annu. Rev. Entomol.* **62**, 443-460 (2017).
- Andersson, M. *et al.* Carbonic anhydrase generates CO² and H⁺ that drive spider silk formation via

- opposite effects on the terminal domains. *PLoS Biol.* **12**, e1001921 (2014).
8. Andersson, M., Holm, L., Ridderstrale, Y., Johansson, J. & Rising, A. Morphology and composition of the spider major ampullate gland and dragline silk. *Biomacromolecules* **14**, 2945-52 (2013).
 9. Knight, D.P. & Vollrath, F. Changes in element composition along the spinning duct in a *Nephila* spider. *Naturwissenschaften* **88**, 179-82 (2001).
 10. Xu, M. & Lewis, R.V. Structure of a protein superfiber: spider dragline silk. *Proc. Natl. Acad. Sci. USA* **87**, 7120-4 (1990).
 11. Rising, A. *et al.* Spider silk proteins--mechanical property and gene sequence. *Zoolog Sci* **22**, 273-81 (2005).
 12. Stellwagen, S.D. & Renberg, R.L. Toward Spider Glue: Long Read Scaffolding for Extreme Length and Repetitious Silk Family Genes AgSp1 and AgSp2 with Insights into Functional Adaptation. *G3 (Bethesda)* **9**, 1909-1919 (2019).
 13. Domigan, L.J. *et al.* Carbonic anhydrase generates a pH gradient in *Bombyx mori* silk glands. *Insect Biochem. Molec.* **65**, 100-6 (2015).
 14. Zhou, L., Chen, X., Shao, Z., Huang, Y. & Knight, D.P. Effect of metallic ions on silk formation in the Mulberry silkworm, *Bombyx mori*. *J. Phys. Chem. B.* **109**, 16937-45 (2005).
 15. Gronau, G., Qin, Z. & Buehler, M.J. Effect of sodium chloride on the structure and stability of spider silk's N-terminal protein domain. *Biomater Sci.* **1**, 276-284 (2013).
 16. Wang, X. *et al.* Chitin and cuticle proteins form the cuticular layer in the spinning duct of silkworm. *Acta Biomater.* **145**, 260-271 (2022).
 17. Davies, G.J., Knight, D.P. & Vollrath, F. Chitin in the silk gland ducts of the spider *Nephila edulis* and the silkworm *Bombyx mori*. *PLoS One* **8**, e73225 (2013).
 18. Salmena, L., Poliseno, L., Tay, Y., Kats, L. & Pandolfi, P.P. A ceRNA hypothesis: the Rosetta Stone of a hidden RNA language? *Cell* **146**, 353-8 (2011).
 19. Kuhn, D.E. *et al.* Experimental validation of miRNA targets. *Methods.* **44**, 47-54 (2008).
 20. Kruger, J. & Rehmsmeier, M. RNAhybrid: microRNA target prediction easy, fast and flexible. *Nucleic Acids Res.* **34**, W451-4 (2006).
 21. Shannon, P. *et al.* Cytoscape: a software environment for integrated models of biomolecular interaction networks. *Genome Res.* **13**, 2498-504 (2003).
 22. Dicko, C., Vollrath, F. & Kenney, J.M. Spider silk protein refolding is controlled by changing pH.

Biomacromolecules **5**, 704-10 (2004).

23. Buchberger, A., Bukau, B. & Sommer, T. Protein quality control in the cytosol and the endoplasmic reticulum: brothers in arms. *Mol Cell* **40**, 238-52 (2010).

24. Ellgaard, L. & Helenius, A. Quality control in the endoplasmic reticulum. *Nat. Rev. Mol. Cell Biol.* **4**, 181-91 (2003).

Reviewers' Comments:

Reviewer #2:

Remarks to the Author:

I appreciate the thoroughness of the author's revisions. My concerns with the prior draft of the manuscript have been addressed.

Reviewer #3:

Remarks to the Author:

I am happy to see the new version of the manuscript. I suggest accept the manuscript for publishing.

RESPONSE TO REVIEWER COMMENTS:

We thank the Reviewers for the time and effort spent carefully reviewing our manuscript and providing constructive comments. Point-by-point responses to all comments and modifications to the manuscript are listed below. The reviewers' comments are in plain text, and our responses are in blue. The cross-references to the manuscript are bold and underlined>. (Line numbers mentioned in the responses may not coincide with the original line numbers.)

Reviewer #1 (Remarks to the Author):

In this study, the authors performed a molecular atlas for the major ampullate (Ma) gland of the golden orb-web spider *Trichonephila clavata*. As described in the text, this atlas comprises a chromosome-scale reference genome and multiomics data, including RNA sequencing (RNA-seq), liquid chromatography–mass spectrometry (LC–MS), the assay for transposase-accessible chromatin with sequencing (ATAC-seq), bisulfite-seq, whole-transcriptomics (WT), single-cell (SC) RNA-seq, and spatial transcriptomics (ST) data. On the basis of these data, the authors revealed a hierarchical biosynthesis of spidroins, organic acids, lipids, and chitin in the sectionalized Ma gland. To probe the cellular characteristics of Ma gland, the authors performed SC RNA-seq analysis and identified ten cell types in the Ma gland. Generally, this study provided new data and insights into the Ma gland of *T. clavata*, which is of potential value. However, I had expected much more than what the manuscript told at this step.

This study attempt to generate a molecular atlas for the Ma gland of *T. clavata*, thereby elucidating the tri-sectional spinning mechanism of spider dragline silk. This could be an interesting study, however, there have been a lot of papers describing the genome and/or transcriptome of spider species and many interesting genes revealed based on these data, I did not find this study to make a more valuable contribution to the literature or to science.

First of all, it is obvious that the authors have read a number of literature related to the issues they discuss, but it is uncertain about the novelty of the scientific questions to be solved by this study. What is impressive in this study is that the authors analyzed the Ma gland of the *T. clavata* through chromosome-scale reference genome assembly and several sequencing technologies such as SC and ST RNA-seq. Of course, there is no doubt that all sequencing data have their own scientific value.

Furthermore, the whole manuscript is too much descriptive, with so many trivial findings in the text but without compelling highlights or emphasis, it is very hard to catch the novel point(s) from the current manuscript. In this study, both the chromosome-scale reference genome assembly and single-cell analysis on the Ma gland of *T. clavata* are actually interesting and valuable. The authors have attempted to describe the results of data analysis, including the expression of some representative genes, however, each part seems to be insufficient to reveal the key scientific questions. In depth analysis and necessary experimental verification are needed.

In summary, this work provided some new data and preliminary analysis on the Ma gland of *T. clavata*, it seems be more suitable for publication in the professional journals of data analysis.

Response: We thank the reviewer for the time and effort that you have put into reviewing the previous version of our manuscript. We appreciate the detailed and constructive comments. These comments are all valuable and very helpful for revising and improving our paper, as well as the important guiding significance to our research. We have studied the comments carefully and made corrections that we hope meet with approval.

The process of natural silk production in the spider major ampullate (Ma) gland endows dragline silk with extraordinary mechanical properties and the potential for biomimetic applications. Although many papers have described the genome and transcriptome of spider species and revealed many interesting genes based on these data¹⁻⁴, the precise genetic roles of the tri-sectional Ma gland during the silk-production process remain unknown. Most studies on the Ma gland tend to treat it as a whole rather than discuss it in the Tail, Sac, and Duct segments. Much of our current understanding of these segments is based on physical and material studies that have provided only a partial picture of their nature^{5,6}. For example, they are characterized by gradients of pH values, ion concentrations, and shear forces⁷⁻⁹. Therefore, it is necessary to divide the Ma gland into Tail, Sac, and Duct to explain the molecular biological mechanism of dragline silk formation. Our work performed the first molecular atlas of natural spider dragline silk production using genome assembly for the golden orb-web spider *Trichonephila clavata* and multiomics defining for the segmented major ampullate (Ma) gland: Tail, Sac, and Duct, which showed the precise gene regulation, protein secretion, cell architecture, and anatomic form of the Ma gland segments. The genetically mediated mechanisms of dragline silk production in the Ma gland will facilitate an understanding of the spider silk-spinning process and guide significance

for optimizing artificial spider silk and establishing a biomimetic system.

We agree with your valuable feedback that our research has so many trivial findings and the novelty of the scientific questions to be solved by this study is uncertain. As you said, some sections of this paper do have some shortcomings. Therefore, we have optimized our manuscript writing, conducted in-depth data analysis, made relevant corrections, and summarized the highlights of our study compared with previous spider genomic studies. This revised paper is tightly focused on the mechanism of dragline silk production and stands out in the following aspects:

(1). A high-quality spider genome is the cornerstone of the whole research. The *T. clavata* genome was assembled to the chromosome level by using the most comprehensive sequencing data (Supplementary Data 1). These data included 129 × Nanopore reads, 73 × Illumina reads, 161 × Hi-C reads, the transcriptome of 18 tissues (three replicates for each sample), and PacBio Iso-seq data (two libraries: 0–10 k and 5–10 k). Based on these data, we assembled a 2.72 Gb high-quality *T. clavata* genome with a heterozygosity of 1.44%. We found that DNA transposons and LTR retrotransposons were responsible for the enlargement of the *T. clavata* genome (Supplementary Fig. 7). Moreover, we also assembled highly repetitive spidroin sequences and found that MaSps and MiSps were clustered on chromosomes (Fig. 1c). The evolutionary analysis indicated that MaSp groups 1 and 2 have a recent common ancestral origin between species (Supplementary Fig. 12). These results provide a solid foundation for the paper and an important reference for subsequent spider-related studies.

(2). Multiomics analysis defines the proteomic and metabolic components of dragline silk and traces their origins from the tri-sectional Ma gland. Given that we are captivated by the incomparable performance of spider dragline silk, our goals are to comprehensively explore the substances and genes related to the synthesis of dragline silk and provide a reference for artificial spinning. Most studies have focused on some particular spidroins, which lack a comprehensive and in-depth study of dragline silk. This paper makes up for this shortcoming and elaborates the components of dragline silk and the silk production mechanism of the Ma gland in detail by using multiple omics data. The present study is the first to divide the spider Ma gland into three sections for multiomics analysis. Our multiomics analysis revealed the highlighted results:

(i). The MaSp content was the highest in dragline silk, and the organic acid content was the richest in the metabolites (Fig. 2b, c).

(ii). The synthesis of dragline silk was closely related to the organic acids of Tail, the lipids of Sac, and

the chitin of Duct (Fig. 2j).

(iii). The differential expression pattern of the MaSp groups was related to epigenetic modification and transcriptional regulation (Fig. 3).

(iv). The rich MaSp content in dragline silk was due to a large number of MaSp secretory cells in the Ma silk gland, and these cells were mainly concentrated in the Tail and Sac.

(v). We also characterized the spatial expression positions of genes related to organic acids, lipids, ions, and chitin in the silk glands, further explaining the rational and orderly synthesis of silk proteins in the silk gland lumen.

We believe that these characteristics can provide a new understanding of dragline silk production and are of guiding significance for the modification of dragline silks and artificial spider silk in vitro.

(3). The first single-cell spatial architecture at the whole-Ma-gland scale. We strongly agree that the single-cell (SC) RNA-seq and spatial transcriptomics (ST) data you mentioned are of scientific significance. For spiders, this is the first single-cell map of the spider's Ma gland. We used SC and ST data to identify cell types and spatial distribution and carried out various available analyses, including cell cluster classification, functional annotation, and pseudotime analysis. This greatly broadens the scope of spider research. Future researchers can take this map as a reference to carry out the precise transformation, development, and utilization of higher-value spider silk, etc., which is of great significance for the vertical and horizontal expansion of spider silk production. According to the single-cell expression profiles, we divided them into 10 cell clusters, which further verified the difference between the Tail, Sac, and Duct of the Ma gland. We found that the cell types with different signature genes were distributed in the Tail, Sac, and Duct regions, playing different functions to coordinate the production of dragline silk. Combined with ST data, it was further found that Sac and Duct were derived from the early cell types existing in the Tail through cell differentiation. Whether using multiomics data or single-cell RNA-seq data, we all found a difference between the Tail, Sac, and Duct of the Ma gland, which proved the necessity and significance of the analysis according to the Tail, Sac, and Duct structures.

(4). Molecular analogs between the spider Ma gland and silkworm silk gland for silk formation. We considered the silkworm silk gland as a bioreactor for the synthesis of spider silk. Therefore, we performed a multiomics comparison and analysis of silk and silk glands between silkworm and spider and found many convergent and similar substances and genes. However, due to the lack of functional

research on many spider genes and the lack of a stable genetic system, we cannot verify the function of genes through experiments, which is truly a pity. To overcome these difficulties, we verified the spinning mechanism of the Tail, Sac, and Duct in the Ma gland through the circuitous way of silkworms through genetic manipulation experiments and proved the possibility of the theory to a certain extent. The analogous molecular characteristics listed here further suggested another important referable method, silkworm-silk-gland-bioreactor, for spider silk synthesis.

(i). Both silkworms and spiders had similarities in the morphology and function of their silk glands. The posterior part mainly secretes silk proteins, the middle part is enriched in lipid-related terms, and the anterior part is enriched in ion- and chitin-related terms (Fig. 5a, b, and f).

(ii). There were a large number of genes with homologous and consistent expression trends in the silk gland tissues of silkworm and spider.

(iii). Silkworm CRISPR/Cas9 experiments also verified that spider homology genes were closely related to the yield of silk synthesis (Fig. 5d, e).

We also thank the Reviewer for the comments concerning our scientific writing. Indeed, we agree that the previous version of our manuscript was too descriptive, and it was difficult to capture the novel points. As advised, to focus on the mechanism of dragline silk production and stand out the above aspects, we have more accurately optimized the background, purpose, and significance of this study to the introduction section, summarized and condensed our results, added in-depth analyses to the results section, and added relevant notes to the discussion section of the manuscript. The revised portions are marked in blue in the revised paper.

In summary, our research is extensive and profound in the molecular biological mechanism of dragline silk production. The results of the *T. clavata* high-quality genome, Ma gland multiomics characteristics, and comparative analysis for silk glands between silkworm and spider expands the knowledge of spider silk production and may eventually benefit spider-inspired fiber innovations. In addition, we are concerned that the data value of this study will be overwhelmed and then developed the SpiderDB database (<https://spider.bioinfotoolkits.net>). The database was an open and accessible online resource for presenting multiomics data of *T. clavata* in an interactive user interface so that interested researchers could further study. Thank you again for your suggestions, and we hope to learn more from you.

Reviewer #2 (Remarks to the Author):

Hu et al. have put together a remarkably thorough investigation of the *Trichonephila clavata* genome and the major ampullate gland molecular biology. Quite impressive! Of particular note is the apparent high quality of the genome and the multi-layered approach to the gene expression and potential regulation in not only the whole major ampullate gland, but different sections and even determining individual cell types. Prior to publications, I'd like to see some clarifications and revisions.

Main concerns

1. Spidroins

a. Identification – I am worried about the inclusion of spidroins that lack a N and C-terminal domain, especially regarding ones with no previously annotated repeat structure or motifs. These would include Sp-AGL-rich, Sp-GP-rich. In general, the Sp... spidroins are not convincing based on motifs or expression patterns. The MaSp9 motif class reported in supplementary table S8 is not sufficient to convince me this is a spidroin. AgSp3 and AgSp4 motifs do look like AgSp1 and AgSp2 but Other proteins can generate similar sequences. The PySp motifs are not particularly convincing either. Significant BLASTp alignments of AgSp3, AgSp4 and PySp to the homologous spidroins in a search against NCBI nr might be more convincing.

Response: We thank the Reviewer for the comments and agree with the suggestions. We apologize for our inaccurate description of spidroins that lack N- and C-terminal domains. In fact, the “spidroins that lack N- and C-terminal domains” indicate that the spidroins lack typical/conserved N- and C-terminal domains. Among the 28 spidroins we identified, 26 have nonrepetitive N- and C-terminal domains, and 15 have typical/conserved N- and C-terminal domains. We have corrected Supplementary Fig. 11 (Figure R1) and rewritten the relevant descriptive sentence in the manuscript.

Figure R1: Protein structure of 28 spidroins. Green and orange boxes indicate the typical/conserved N- and C-terminal domains of spidroins. Gray boxes indicate the atypical N- and C-terminal domains.

As advised, we downloaded 433 redundant spidroin sequences from the NCBI protein database (<https://www.ncbi.nlm.nih.gov/protein>). The protein sequences of AgSp3, AgSp4, PySp, Sp-AGL-rich, and Sp-GP-rich were used to align these spidroins by blastp (E-value < 1e⁻⁵). As shown in Figure R2a-d, we generated network diagrams of AgSp3 (renamed AgSp3a), AgSp4 (renamed AgSp2), MaSp9 (renamed MaSp-like), Sp-AGL-rich, Sp-GP-rich, and PySp and found that all of them were aligned to the reference spidroin sequences, suggesting that they are spidroin-like proteins.

MaSp9 (renamed MaSp-like) has nonrepetitive N&C-terminal domains and a repetitive domain (Figure R2e). The sequences of GGX, (GA)_n, and (A)_n are typical motifs in MaSp^{10,11}. We found that the repetitive domain of MaSp9 lacked the (A)_n motif but was rich in GGX and (GA)_n motifs. Therefore, we redefined the MaSp9 protein as a MaSp-like protein.

Figure R2: **a-d**, Network diagram of AgSp2 (a), AgSp3a (a), PySp (b), Sp-AGL-rich (c), and Sp-GP-rich (d). Red circles indicate the query protein, and other circles indicate reference spidroins. **e**, Protein sequence of MaSp-like showing GGX- and $(GA)_n$ -rich motifs.

b. Phylogenetic analysis – it looks like full-length sequences were used. The repetitive structure of spidroins makes interpreting phylogenetic reconstruction of full-length sequences extremely challenging. In fact, the repeat sequences shouldn't even align across many types. I strongly recommend limiting phylogenetic analysis to the N and C-terminal domains. If this was done, please clarify.

Response: We thank the Reviewer for the comments and agree with the suggestions. In the manuscript, we used the full-length sequences to construct the phylogenetic tree due to the lack of a nonrepetitive

N/C-terminal domain in two spidroins (AgSp2 and AgSp3b). However, it was indeed challenging. As you suggested, we reconstructed the phylogenetic tree of spidroins using the N-terminal domain because the N-terminal domain has a longer sequence than the C-terminal domain. We have corrected Supplementary Fig. 12a and renamed all spidroins depending on this phylogenetic tree (Figure R3).

Figure R3: Phylogenetic analysis of 28 spidroins. The BmFibH of *Bombyx mori* is an out-group.

c. Naming – by convention MaSp1 has represented spidroins with GGX and poly-A motifs, and MaSp2 also included GPGXX. The sequential naming of the MaSp as 1-9 and then calling them group 1 and group 2 is confusing. I admit I’m also stumped as to the best way to name these, but if a MaSp has a motif unit typical of MaSp1, it should be called “MaSp1”. Then to separate the multiple copies, they could be named “MaSp1 locus 1”, “MaSp1 locus 2”, etc. Similar for MaSp2. Some additional MaSp sequences have been identified so to be called MaSp3, etc the MaSp probably needs to have a best BLASTp alignment to one of those alternative spidroins. However, I admit that there is not a clear convention past MaSp1 and MaSp2.

Response: We thank the Reviewer for the comments and agree with the suggestions. Indeed, we are stumped by the spidroin naming. According to the suggestions, we have renamed the MaSpS that had GGX and (A)_n motifs “MaSp1” and that had GPGXX, GGX, and (A)_n motifs “MaSp2”. The subclassification

of MaSp1s (MaSp1a–c) and MaSp2s (MaSp2a–e) was performed by sequence alignment and phylogenetic analysis using the N terminal domain (Figure R4).

Figure R4: **a**, Motif analysis of MaSp. Yellow boxes indicate proteins containing the corresponding motif. **b**, Phylogenetic analysis of MaSp.

d. Naming AgSp – Similar comment where AgSp1 has a very different repeat organization than AgSp2 as defined in Stellwagen & Renberg (G3, 2019).

Response: We thank the Reviewer for the suggestion. We downloaded AgSp1 and AgSp2 sequences of *A. trifasciata* reported by Stellwagen & Renberg (G3, 2019)¹². According to the sequence alignment and phylogenetic analysis of AgSp between *T. clavata* and *A. trifasciata*, we renamed the AgSp of *T. clavata* (Figure R5).

Figure R5: **a-b**, Sequence alignment (a) and phylogenetic analysis (b) of AgSp between *T. clavata* and *A. trifasciata*.

2. Convergent evolution

a. While I found the comparisons with Bombyx silk glands interesting, I was not particularly convinced that there was much overlap in molecular functions other than the gross morphology of the major ampullate gland with the Bombyx gland.

Response: We thank the Reviewer for the comment. As the Reviewer said, the major ampullate gland

and the *Bombyx* gland were grossly similar in morphology. We proposed molecular functional similarities between the major ampullate gland and the *Bombyx* gland based on the following points:

(1). Previous studies have found pH gradients in both the major ampullate gland and the *Bombyx* gland, which from base to acid form Tail (7.6)/PSG (8.2) to Duct (5.7)/ASG (6.2)^{7,8,13}. Carbonic anhydrase was found to be responsible for the generation and maintenance of the pH gradient of both the major ampullate gland and the *Bombyx* gland^{7,8,13}. The pH gradients have been inferred to be important for fiber formation⁸.

(2). Previous studies have found ion concentration gradients in both the major ampullate gland and the *Bombyx* gland. For example, the sodium levels were found to increase from the distal to proximal parts of both the major ampullate gland and *Bombyx* gland using X-ray emission^{9,14}. An increased concentration of sodium chloride is important in destabilizing the silk protein homodimer¹⁵.

(3). Previous studies have found chitin in the cuticle of the spinning ducts of both the major ampullate gland and the *Bombyx* gland^{16,17}. Chitin is an essential component for the construction of spinning ducts and for generating shear forces^{8,16,17}

(4). Our results found similarities between the major ampullate gland and the *Bombyx* gland, including calcium ion binding, chitin binding, signal transduction, and *V-type H⁺-transporting ATPase* expression. Furthermore, we identified convergent silk components in spider and silkworm silks, including two proteins (mucin-19 and GDH) and two major metabolites (choline and DL-malic acid). Some of the analogous aspects have been reported previously, and some of them were novelty identified, which needed to be further studied.

In summary, we propose that there are molecular functional similarities between the major ampullate gland and the *Bombyx* gland in the silk formation processes of the silk gland.

b. My main concern is that I suspect if any two tissue types were compared across species, some overlap would be found, especially for tissues that are likely to be expressing a lot of extruded protein. Thus, I'd like to see comparisons between other tissues to exclude the possibility that there is simply going to be overlap if you dig deep enough.

Response: We thank the Reviewer for the comment. As advised, we have added two other *T. clavata* tissues (hemocyte and ovary) to the Venn comparison analysis. We identified 5,319 and 2,222 tissue-

specific expressed genes in hemocyte and ovary, respectively (Figure R6a-b). Gene Ontology (GO) analysis identified 354 and 67 GO terms enriched in hemocyte and ovary, respectively (Figure R6c-d). The shared GO terms of different tissues indicated the analogous aspects in tissue-developing and function-performing. As shown in Figure R6c, no GO term was identified in the comparison groups of Hemocyte vs. Tail, Hemocyte vs. Sac, Hemocyte vs. PSG, and Hemocyte vs. ASG. Only three shared GO terms (cytoskeleton organization, cytoskeletal protein binding, and GTPase activity) and one shared GO term (NAD binding) were identified in the hemocyte vs. Duct and hemocyte vs. MSG comparison groups, respectively. As shown in Figure R6d, no GO term was identified in the comparison groups of Ovary vs. Tail, Ovary vs. Sac, Ovary vs. PSG, Ovary vs. MSG, and Ovary vs. ASG. Only three shared GO terms (microtubule-based process, cytoskeleton organization, and supramolecular complex) were identified in the comparison of Ovary vs. Duct. Our results showed that the overlap analysis in different comparison groups exhibited differentiated GO terms. The shared GO terms between spider and silkworm silk glands were specific and not identified when we compared silk glands with other tissue types (hemocyte and ovary).

Figure R6: **a-b**, Tissue-specific gene expression analysis of the *T. clavata* hemocyte (a) and ovary (b). **c-d**, GO enrichment and Venn comparison analyses of hemocyte vs. silk gland (c) and ovary vs. silk gland (d). A *P* value < 0.05 was set as the criterion for screening enriched GO terms.

Additional concerns, in order of encounter not importance.

3. Write out abbreviations on first use.

Response: We thank the Reviewer for the suggestion. We have checked the full manuscript and written

out abbreviations on first use.

4. “Spiders are a group” should be reworded. “Spiders form a group” would be OK. Or “Spiders (Order Araneae) are abundant generalist...”

Response: We thank the Reviewer for the suggestion. We have reworded this sentence to **“Spiders (Order Araneae) are abundant generalist arthropod predators, including more than fifty thousand extant species.”**

5. Reference numbering is not correct for many references. I am not sure where they go wrong but definitely the references in the second paragraph of the introduction cannot support the statements made in the paragraph. For instance, #22 and 23 are not for recombinant spider papers, and the SpiCE references are #34 and 35 in the bibliography but 24,25 are included in the text. There were many other instances throughout the manuscript where I couldn't find the reference that supported the statement made. These need to be updated throughout.

Response: We thank the Reviewer for the suggestion. We have corrected the incorrect reference number, added appropriate reference citations to support the statement made, and updated all references throughout the manuscript.

6. End of second paragraph of introduction – clarify which aspects of Ma gland biology are poorly understood. I would argue much is already known, although this manuscript certainly expands that knowledge base.

Response: We thank the Reviewer for the suggestion. We agree with the Reviewer's argument and have rewritten the sentence to **“It is becoming apparent that the detailed mechanisms underlying dragline silk production, including the cellular architecture and molecular function of the Ma gland as well as the biocomposition and formation of dragline silk, are fundamental to advanced fiber innovation.”**

7. Line 57: change exhibited and constructed to present tense, since this species has not gone extinct.

Response: We thank the Reviewer for the suggestion. We have corrected the verb “exhibited” to “exhibits”.

8. Line 100: I think it is OK to use homologous or perhaps syntenic to refer to the similar grouping of MaSp sequences on chromosomes across species.

Response: We thank the Reviewer for the suggestion. We have corrected the adjective “analogous” to “homologous” and used “homologous” to refer to the similar grouping of MaSp sequences on chromosomes across species.

9. For any enrichment analysis, please clarify the comparison group.

Response: We thank the Reviewer for the suggestion. We have clarified the comparison group for each enrichment analysis.

a. For instance, line 168... GO terms significantly enriched in the tail... relative to all GO terms assigned to whole genome? Or something else?

Response: The GO terms significantly enriched in the Tail were relative to the GO terms enriched in the other two segments of the Ma gland, the Duct and Sac. We have rewritten the sentence to “**We found that the GO terms that were significantly enriched in the Tail (relative to the Duct and Sac) were mainly related to the synthesis of organic acids (the largest group in the silk metabolome), those in the Sac (relative to the Duct and Tail) were mainly related to the synthesis of lipids (the third-largest group in the silk metabolome), and those in the Duct (relative to the Sac and Tail) were related to ion (Ca²⁺ and H⁺) exchange and chitin synthesis (Fig. 2j).**”

b. As another instance, line 198... what is enrichment of TF motifs relative to? Are these just the TFs found in the ATAC-seq peak set, and then compared to all TF motifs? Or the ones in the 2kb upstream

region that aren't in the ATAC-seq peak set?

Response: The “enriched TF motifs” in the manuscript were predicted TF binding sites with E-values $< 1e^{-10}$ when compared the target peaks regions with the genome-wide peaks regions for each tissue. For example, we want to obtain all enriched TF motifs in the upstream and downstream 2k ranges of genome genes in Duct tissue, and we compared enriched TF motifs in these chromatin open regions with enriched TF motifs in the chromatin open regions of the entire genome. We apologize for our confusing expression. We have corrected “enriched TF motifs” to “potential TF motifs” and rewritten the sentence to **“We analyzed potential TF motifs among the ATAC-seq peak sets in the 2 kb regions upstream of the transcriptional start sites (TSSs).”** The “enriched TF motifs” in the following sentences and Figure 3 legend have also been corrected.

10. Lines 184-188: I'd like a little more context for the mean RPKM values reported for chromatin accessibility. The means seem to swamp the higher signal of accessibility surrounding the transcription start sites (TSS) as shown in Figure 3a. Specifically, the peak of RPKM looks similar among the 3 regions of the MA gland, so the conclusion that the duct has low accessibility isn't entirely clear to me. Would describing the number of genes with greater than background RPKM for each section help? Or the extent of the accessibility? E.g. text that explains the pattern in Figure 3a might be more intuitive?

Response: We thank the Reviewer for the suggestion. As advised, we added two sets of data to illustrate the chromatin accessibility (CA) of each section. We counted the number of significant ATAC peaks with greater than background RPKM for each silk gland segment. **“A total of 702,037 (Tail), 767,517 (Sac), and 653,361 (Duct) significant ATAC peaks (RPKM > 2) were identified in the 2 kb regions upstream and downstream of genes, and 10,501,151 (Tail), 11,356,55 (Sac), and 9,778,368 (Duct) significant ATAC peaks (RPKM > 2) were identified at the whole-genome level.”** These results suggested that the duct has relatively low accessibility compared with the Tail and Sac. We have added an explanation of Figure 3a in the legend and added the comparison of the number of significant ATAC peaks in the manuscript.

11. I have to admit my own ignorance that I had never seen the term ceRNA before. I think it would be

useful to add a brief explanation of context to the results and first mention of ceRNA. E.g. are there really mRNAs that compete for microRNA regulation? What would this look like and why would it matter for dragline silk gene expression, regulation, silk formation? The methods for identifying the network in Figure 3g is not clear to me from the results, methods, or the supplementary note. Probably just some context language around the goals of miRanda and RNAhybrid would help here. These look like programs that identify miRNA binding sites. Was the network, just build by hand after that? Or do these programs also implement network algorithms?

Response: We thank the Reviewer for the suggestion. Based on the suggestions, we have rewritten the sentences in the ceRNA section as follows: 1) We added a brief explanation of the context of the first mention of ceRNA in the manuscript: **“To investigate the impact of competing endogenous RNAs (ceRNAs: a post-transcriptional regulatory system implemented by miRNA and lncRNA¹⁸) corresponding to the regulation of dragline silk genes, we performed a whole-transcriptomic analysis of the tri-sectional_Ma gland...”**. 2) We added the expression pattern of the lncRNA and miRNA to explain the potential regulation pattern of the lncRNA–miRNA–mRNA competition network for dragline silk gene expression: **“Remarkably, we noted that the ceRNA networks of *MaSp1a–c* & *MaSp2e* (*MaSp*-Group 1) were tightly clustered, as were those of *MaSp2a–d* (*MaSp*-Group 2); three lncRNAs (LXLOC_047988, LXLOC_047990, and LXLOC_051464) in the *MaSp*-Group 1 network were highly expressed in the Ma gland (Supplementary Fig. 18); one lncRNA (LXLOC_070389) and four miRNAs (novel mir42, novel mir46, novel mir166, and miR-285 3) in the *MaSp*-Group 2 network were highly expressed in the Ma gland (Supplementary Fig. 18); in addition, the ceRNA networks of the two *MaSp* groups were independent of each other (Fig. 3g; Supplementary Fig. 18). These results further revealed potential post-transcriptional networks and the differentiated coregulatory pattern of the genes in the two *MaSp* groups in the Ma gland”**. 3) We added some context language around the goals of miRanda and RNAhybrid: **“From these data, we constructed potential lncRNA–miRNA–mRNA interaction pairs by using the miRanda¹⁹ and RNAhybrid²⁰ algorithms to identify the potential binding site between miRNA and lncRNA/mRNA, and then”**. 4) We added an explanation of the software used to construct and visualize networks: **“and then visualized the interaction networks by using Cytoscape software^{21”}**.

12. After teaching myself about ceRNA and figuring out what the network in Fig 3g is showing, I'm not convinced there is much opportunity for competition. I'm more convinced of the potential for coregulation of MaSp-1 and coregulation of MaSp-2. Again, context and clarification would help.

Response: We thank the Reviewer for the suggestion. We found that the mRNAs of *MaSp1a-c* & *MaSp2e* (*MaSp*-Group 1) tightly interacted with seven lncRNAs and two miRNAs, and the mRNA of *MaSp2a-d* (*MaSp*-Group 2) tightly interacted with four lncRNAs and four miRNAs (Fig. 3g). Among them, three lncRNAs (LXLOC_047988, LXLOC_047990, and LXLOC_051464) in the *MaSp*-Group 1 network were highly expressed in the Ma gland (Supplementary Fig. 18); one lncRNA (LXLOC_070389) and four miRNAs (novel_mir42, novel_mir46, novel_mir166, and miR-285_3) in the *MaSp*-Group 2 network were highly expressed in the Ma gland (Supplementary Fig. 18). Therefore, we suspected a potential regulatory pattern of the lncRNA-miRNA-mRNA competition network for *MaSp*-Group 1 and *MaSp*-Group 2 gene expression. However, their regulation patterns were independent of each other. As advised, we have added the context and clarification to the coregulation of *MaSp*-Group 1 and the coregulation of *MaSp*-Group 2: **"These results further revealed potential post-transcriptional networks and the differentiated coregulatory pattern of the genes in the two *MaSp* groups in the Ma gland"**.

13. Line 209: Define WT as Whole Transcriptome.

Response: We thank the Reviewer for the suggestion. To avoid the confusing use of abbreviations between whole-transcriptome and wild type, we have defined WT as wild type and removed all of the abbreviations for whole-transcriptome in the manuscript.

14. Line 341: Prior work established importance of duct in protein structural transition. Please cite it here.

Response: We thank the Reviewer for the suggestion. We have cited relevant references on the importance of Duct in protein structural transition.

15. Line 358: “shared” should be “shear” (I think)

Response: We thank the Reviewer for the suggestion. We have corrected “shared forces” to “shear forces”.

16. Line 370: What is SC and ST?

Response: We have defined abbreviations SC as single-cell and ST as spatial transcriptomics in the manuscript.

17. Line 371: discuss meaning of networks rather than relying on reader to intuit ceRNA

Response: We thank the Reviewer for the suggestion. We have added the relevant discussion to the meaning of ceRNA networks and their potential regulation pattern: “Our results provide genomic clues for the hierarchically ordered biosynthesis of spidroins. We documented that the *MaSp1a-c* & *MaSp2e*, *MaSp2a-d*, and *MiSp-a-e* genes are distributed in three distinct groups. In addition, we demonstrated that the *MaSps* within each of these groups exhibited concerted SC and ST expression profiles in the tri-sectional Ma gland. We also identified the group-specific common TF motifs at the epigenetic level and constructed group-specific lncRNA-miRNA-mRNA networks at the ceRNA level. Such results revealed novel structural, expressional, and regulatory characteristics of spidroin genes that have not been reported in other spider genomes.”

18. Lines 380-381: What is the debate regarding the Ma cell types? How does your work support one side or another, on no past work?

Response: The number of cell types in the Ma gland has been a subject of debate. Based on tissue sectioning, staining, and optical observations, previous studies have shown that there are two or three cell types in the Ma gland of orb-web spiders^{8,22}. However, our study provides single-cell sequencing data indicating the existence of ten cell types in the whole Ma gland, which supports no past work.

19. Lines 389-401: As mentioned above, I'm not convinced the analogous expression between spider MA and bombyx silk glands is that meaningful for silk production. Might want to remove this section, pull back on conclusions, or need to convince reader that other tissue types wouldn't have some overlap too.

Response: We thank the Reviewer for the suggestion. We have performed two *T. clavata* tissues (hemocyte and ovary) in the Venn comparison analysis. As shown in Figure R6a-b, no GO term was identified in the comparison groups of Hemocyte vs. Tail, Hemocyte vs. Sac, Hemocyte vs. PSG, Hemocyte vs. ASG, Ovary vs. Tail, Ovary vs. Sac, Ovary vs. PSG, Ovary vs. MSG, and Ovary vs. ASG. Three, one, and three shared GO terms were identified in the hemocyte vs. Duct, hemocyte vs. MSG, and Ovary vs. Duct comparison groups, respectively. More importantly, the overlapping GO terms in different comparison groups were differentiated. The shared GO terms indicated the analogous aspects in the tissue developing and performing functions. In summary, the nonsilk gland tissue types did not have any overlap, which was consistent with the comparison group of spider Ma vs. *Bombyx* silk glands. We have added necessary notes to the discussion section of the manuscript: **“Our work provides biological evidence of this role of the silk gland and further demonstrates high convergence of several molecular functions in the spider Ma gland Duct and the silkworm ASG, including calcium ion binding, chitin binding, signal transduction, and V-type H⁺-transporting ATPase expression. Such overlaps were silk gland-specific and not identified in other tissue types (hemocyte and ovary) (Supplementary Fig. 30).”**

20. Line 483: should “1,5000 x g” be “1,500 x g” or “15,000 x g”?

Response: We thank the Reviewer for the suggestion. We have corrected “1,5000 x g” to **“15,000 x g”**.

21. Line 501: there seems to be a word missing in “were by determining”. Maybe “were estimated by determining”

Response: We thank the Reviewer for the suggestion. We have added the missing word **“estimated”**.

22. Figure 4 legend: I think white-filled cycles and black-filled cycles should be circles instead of cycles. Also, what are the units of expression for part e?

Response: We thank the Reviewer for the suggestion. We have corrected “white-filled cycles” and “black-filled cycles” to “white-filled circles” and “black-filled circles”, respectively. Additionally, we have added the unit of expression, $\text{Log}_2(\text{transcripts per kilobase million; TPM})$, for part e in Figure 4e.

Reviewer #3 (Remarks to the Author):

This manuscript reports the (near) chromosomal level assembly of a golden orb-weaving spider *Trichonephila clavata* with the high-quality. The authors focus on molecular model analyses of spider silk gland (Ma silk gland). The comprehensive multiomics studies and applications are performed in this manuscript, showing their huge workload in terms of sampling and bioinformatics analyses. Such detailed biological mechanism of silk is an important foundation for future study about silk gland, like in evolution and silk spinning behavior. The presented work is quite extensive and the manuscript is well written and illustrated by clear informative figures.

A few other comments:

1) In major manuscript, authors use fragments per kilobase of transcript per million mapped reads (FPKM) to measure the expression abundance of genes (like line 179-181), but in figure 6, authors use RPKM, which confuses me. As far as I know, RPKM means reads per kilobase of exon model per million mapped reads. I would encourage authors to use same normalization method about RNA-Seq, or give an explanation in the legend.

Response: We thank the Reviewer for the suggestion. We apologize for this mistake, the misuse of RPKM in Figure 6 legend. The FPKM was used to measure the expression abundance of genes. We have corrected “RPKM” to “FPKM” in Figure 6 legend.

2) CRISPR/Cas9-based experiment in the silkworm is executed, and the silk production (cocoon weight

and cocoon layer rate) of the Hsp20-KO strain was significantly lower than those of the wild type. Does this imply that if the gene Hsp20 is knocked out in spiders would it have a similar effect?

Response: We agree with the Reviewer's comments. In cells, the accumulation of damaged proteins, resulting from mutation/misfolding/aggregation, can perturb cellular homeostasis and endanger survival under severe stress conditions^{23,24}. The *Hsp20* gene encodes a small heat shock protein that acts as a protein chaperone to protect other proteins against misfolding and aggregation²³. Based on the following reasons: 1) the silk gland is an organ assembling and secreting large amounts of protein and requires the protein damage protector, 2) the pair of *Hsp20* genes was orthologous with a high sequence identity of 87.1% between *T. clavata* and *B. mori*, 3) the pair of *Hsp20* genes was expressed in the silk glands of both *T. clavata* and *B. mori*, 4) the *Hsp20* gene served as a marker gene of SC cluster 5 of the *T. clavata* Ma gland, and 5) the silkworm *Hsp20*-KO strain presented a decreased silk production ability, we suspect that knocking out *the Hsp20* gene in spiders would cause phenotypes similar to those in silkworms (decreasing silk production).

3) In the part of convergent evolution of the tri-section silk gland between spider and silkworm, tail of spider silk gland and PSG of silk shows no sharing GO terms. Many GO terms are similar in secondary structure even though the GO numbers are different, and I'm worried that more similar information may be missed. In addition, whether there are same transcription factors can be checked in tail of spider silk gland and PSG of silk, based on the numerous orthologs between two species.

Response: We thank the Reviewer for the suggestion. As advised, we have investigated the shared GO term at level 2. As shown in Figure R7a, there were ten shared level 2 GO terms between the silk glands of spider and silkworm, including 1) transporter activity, biological regulation, and binding in the comparison group of Duct vs. ASG; 2) metabolic process, cellular process, catalytic activity, cellular anatomical entity, and binding in the comparison group of Sac vs. MSG; and 3) catalytic activity and metabolic process in the comparison group of Tail vs. PSG.

We also screened the same transcription factors (TFs) and found 285, 257, and 241 expressed orthologous TFs in the Duct & ASG, Sac & MSG, and Tail & PSG groups, respectively (Figure R7b). The top 10 TF types are listed in Figure R7c. We found that three ortholog TF types (zf-C2H2, bZIP, and zf-

NF-X1) were significantly upregulated in both the Duct and ASG (Figure R7d), and none of the ortholog TF types was significantly upregulated in Sac & MSG and Tail & PSG.

Figure R7: **a**, Shared GO terms in the comparison groups of Duct vs. ASG, Sac vs. MSG, and Tail vs. PSG. A P value < 0.05 was set as the criterion for screening enriched GO terms. **b**, Screening of the expressed orthologous TFs in the Duct & ASG, Sac & MSG, and Tail & PSG. **c**, Top 10 TF types in the Duct & ASG, Sac & MSG, and Tail & PSG. **d**, TFs expression convergence between the silk glands of *T. clavata* and *B. mori*.

References

- Babb, P.L. *et al.* The *Nephila clavipes* genome highlights the diversity of spider silk genes and their complex expression. *Nat. Genet.* **49**, 895-903 (2017).
- Kono, N. *et al.* Darwin's bark spider shares a spidroin repertoire with *Caerostris extrusa* but achieves extraordinary silk toughness through gene expression. *Open Biol.* **11**, 210242 (2021).
- Kono, N. *et al.* Orb-weaving spider *Araneus ventricosus* genome elucidates the spidroin gene catalogue. *Sci. Rep.* **9**, 8380 (2019).
- Kono, N. *et al.* Multicomponent nature underlies the extraordinary mechanical properties of spider dragline silk. *Proc. Natl. Acad. Sci. USA* **118**(2021).
- Li, J. *et al.* Spider silk-inspired artificial fibers. *Adv. Sci.* **9**, e2103965 (2022).
- Blamires, S.J., Blackledge, T.A. & Tso, I.M. Physicochemical property variation in spider silk: ecology, evolution, and synthetic production. *Annu. Rev. Entomol.* **62**, 443-460 (2017).
- Andersson, M. *et al.* Carbonic anhydrase generates CO² and H⁺ that drive spider silk formation via

- opposite effects on the terminal domains. *PLoS Biol.* **12**, e1001921 (2014).
8. Andersson, M., Holm, L., Ridderstrale, Y., Johansson, J. & Rising, A. Morphology and composition of the spider major ampullate gland and dragline silk. *Biomacromolecules* **14**, 2945-52 (2013).
 9. Knight, D.P. & Vollrath, F. Changes in element composition along the spinning duct in a *Nephila* spider. *Naturwissenschaften* **88**, 179-82 (2001).
 10. Xu, M. & Lewis, R.V. Structure of a protein superfiber: spider dragline silk. *Proc. Natl. Acad. Sci. USA* **87**, 7120-4 (1990).
 11. Rising, A. *et al.* Spider silk proteins--mechanical property and gene sequence. *Zoolog Sci* **22**, 273-81 (2005).
 12. Stellwagen, S.D. & Renberg, R.L. Toward Spider Glue: Long Read Scaffolding for Extreme Length and Repetitious Silk Family Genes AgSp1 and AgSp2 with Insights into Functional Adaptation. *G3 (Bethesda)* **9**, 1909-1919 (2019).
 13. Domigan, L.J. *et al.* Carbonic anhydrase generates a pH gradient in *Bombyx mori* silk glands. *Insect Biochem. Molec.* **65**, 100-6 (2015).
 14. Zhou, L., Chen, X., Shao, Z., Huang, Y. & Knight, D.P. Effect of metallic ions on silk formation in the Mulberry silkworm, *Bombyx mori*. *J. Phys. Chem. B.* **109**, 16937-45 (2005).
 15. Gronau, G., Qin, Z. & Buehler, M.J. Effect of sodium chloride on the structure and stability of spider silk's N-terminal protein domain. *Biomater Sci.* **1**, 276-284 (2013).
 16. Wang, X. *et al.* Chitin and cuticle proteins form the cuticular layer in the spinning duct of silkworm. *Acta Biomater.* **145**, 260-271 (2022).
 17. Davies, G.J., Knight, D.P. & Vollrath, F. Chitin in the silk gland ducts of the spider *Nephila edulis* and the silkworm *Bombyx mori*. *PLoS One* **8**, e73225 (2013).
 18. Salmena, L., Poliseno, L., Tay, Y., Kats, L. & Pandolfi, P.P. A ceRNA hypothesis: the Rosetta Stone of a hidden RNA language? *Cell* **146**, 353-8 (2011).
 19. Kuhn, D.E. *et al.* Experimental validation of miRNA targets. *Methods.* **44**, 47-54 (2008).
 20. Kruger, J. & Rehmsmeier, M. RNAhybrid: microRNA target prediction easy, fast and flexible. *Nucleic Acids Res.* **34**, W451-4 (2006).
 21. Shannon, P. *et al.* Cytoscape: a software environment for integrated models of biomolecular interaction networks. *Genome Res.* **13**, 2498-504 (2003).
 22. Dicko, C., Vollrath, F. & Kenney, J.M. Spider silk protein refolding is controlled by changing pH.

Biomacromolecules **5**, 704-10 (2004).

23. Buchberger, A., Bukau, B. & Sommer, T. Protein quality control in the cytosol and the endoplasmic reticulum: brothers in arms. *Mol Cell* **40**, 238-52 (2010).

24. Ellgaard, L. & Helenius, A. Quality control in the endoplasmic reticulum. *Nat. Rev. Mol. Cell Biol.* **4**, 181-91 (2003).

RESPONSE TO REVIEWER COMMENTS:

We would like to thank the reviewers for their very constructive comments. This has greatly increased the quality of the paper.

Reviewer #2 (Remarks to the Author):

I appreciate the thoroughness of the author's revisions. My concerns with the prior draft of the manuscript have been addressed.

Response: We are very thankful to the reviewer for all the constructive comments to help us improve our manuscript.

Reviewer #3 (Remarks to the Author):

I am happy to see the new version of the manuscript. I suggest accept the manuscript for publishing.

Response: We thank the reviewer so much for the positive comments.